# Partial Counterfactual Identification of Continuous Outcomes with a Curvature Sensitivity Model

**Valentyn Melnychuk, Dennis Frauen & Stefan Feuerriegel**
LMU Munich & Munich Center for Machine Learning (MCML)
Munich, Germany
`melnychuk@lmu.de`

## Abstract

Counterfactual inference aims to answer retrospective "what if" questions and thus belongs to the most fine-grained type of inference in Pearl's causality ladder. Existing methods for counterfactual inference with continuous outcomes aim at point identification and thus make strong and unnatural assumptions about the underlying structural causal model. In this paper, we relax these assumptions and aim at partial counterfactual identification of continuous outcomes, i.e., when the counterfactual query resides in an ignorance interval with informative bounds. We prove that, in general, the ignorance interval of the counterfactual queries has non-informative bounds, already when functions of structural causal models are continuously differentiable. As a remedy, we propose a novel sensitivity model called *Curvature Sensitivity Model*. This allows us to obtain informative bounds by bounding the curvature of level sets of the functions. We further show that existing point counterfactual identification methods are special cases of our *Curvature Sensitivity Model* when the bound of the curvature is set to zero. We then propose an implementation of our *Curvature Sensitivity Model* in the form of a novel deep generative model, which we call *Augmented Pseudo-Invertible Decoder*. Our implementation employs (i) residual normalizing flows with (ii) variational augmentations. We empirically demonstrate the effectiveness of our *Augmented Pseudo-Invertible Decoder*. To the best of our knowledge, ours is the first partial identification model for Markovian structural causal models with continuous outcomes.

## 1 Introduction

Counterfactual inference aims to answer retrospective "what if" questions. Examples are: *Would a patient's recovery have been faster, had a doctor applied a different treatment? Would my salary be higher, had I studied at a different college?* Counterfactual inference is widely used in data-driven decision-making, such as root cause analysis [18, 139], recommender systems [16, 36, 83], responsibility attribution [50, 73, 77], and personalized medicine [84, 138]. Counterfactual inference is also relevant for various machine learning tasks such as safe policy search [108], reinforcement learning [19, 38, 63, 86, 91], algorithmic fairness [76, 104, 146], and explainability [42, 44, 64, 65, 75].

Counterfactual queries are located at the top of Pearl's ladder of causation [8, 51, 101], i.e., at the third layer $\mathcal{L}_3$ of causation [8] (see Fig. 1, right). Counterfactual queries are challenging as they do reasoning in both the actual world and a hypothetical one where variables are set to different values than they have in reality.

State-of-the-art methods for counterfactual inference typically aim at *point identification*. These works fall into two streams. (1) The first stream [21, 29, 31, 70, 84, 100, 113, 114, 117, 118,

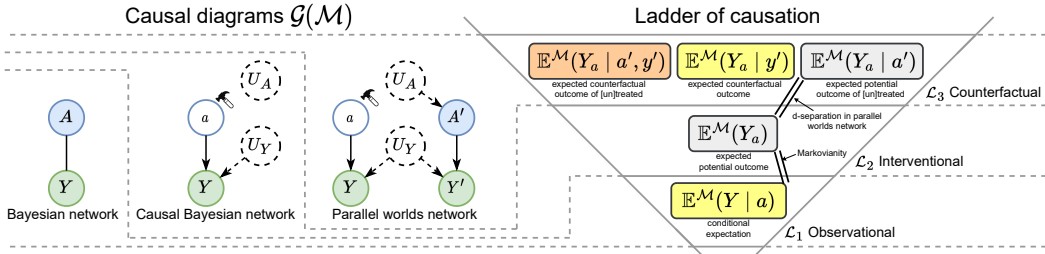

Figure 1: Pearl's ladder of causation [8, 51, 101] comparing observational, interventional, and counterfactual queries corresponding to the SCM $\mathcal{M}$ with two observed variables, i.e., binary treatment $A \in \{0, 1\}$ and continuous outcome $Y \in \mathbb{R}$. We also plot three causal diagrams, $\mathcal{G}(\mathcal{M})$, corresponding to each layer of causation, namely, Bayesian network, causal Bayesian network, and parallel worlds network. Queries with gray background can be simplified, i.e., be expressed via lower-layer distributions. The estimation of the queries with yellow background requires additional assumptions or distributions from the same layer. In this paper, we focus on partial identification of the expected counterfactual outcome of [un]treated, $\mathbb{E}(Y_a \mid a', y'), a' \neq a$, shown in orange.

134, 135, 140] makes no explicit assumptions besides assuming a structural causal model (SCM) with Markovianity (i.e., independence of the latent noise) and thus gives estimates that can be *invalid*. However, additional assumptions are needed in counterfactual inference of layer $\mathcal{L}_3$ to provide identifiability guarantees [94, 137]. (2) The second stream [4, 32, 59, 66, 93, 96, 123, 148] provides such identifiability guarantees but makes strong assumptions that are *unnatural* or *unrealistic*. Formally, the work by Nasr-Esfahany et al. [93] describes bijective generation mechanisms (BGMs), where, in addition to the original Markovianity of SCMs, the functions in the underlying SCMs must be monotonous (strictly increasing or decreasing) with respect to the latent noise. The latter assumption effectively sets the dimensionality of the latent noise to the same dimensionality as the observed (endogenous) variables. However, this is highly unrealistic in real-world settings and is often in violation of domain knowledge. For example, cancer is caused by multiple latent sources of noise (e.g., genes, nutrition, lifestyle, hazardous exposures, and other environmental factors).

In this paper, we depart from point identification for the sake of more general assumptions about both the functions in the SCMs and the latent noise. Instead, we aim at ***partial counterfactual identification*** of continuous outcomes. Rather than inferring a point estimation expression for the counterfactual query, we are interested in inferring a whole *ignorance interval* with *informative bounds*. Informative bounds mean that the ignorance interval is a strict subset of the support of the distribution. The ignorance interval thus contains all possible values of the counterfactual query for SCMs that are consistent with the assumptions and available data. Partial identification is still very useful for decision-making, e.g., when the ignorance interval for a treatment effect is fully below or above zero.

We focus on a Markovian SCM with two observed variables, namely, a binary treatment and a continuous outcome. We consider a causal diagram as in Fig. 1 (left). We then analyze the *expected counterfactual outcome of [un]treated* abbreviated by ECOU [ECOT]. This query is non-trivial in the sense that it can *not* be simplified to a $\mathcal{L}_1/\mathcal{L}_2$ layer, as it requires knowledge about the functions in SCM, but can still be inferred by means of 3-step procedure of abduction-action-prediction [102]. ECOU [ECOT] can be seen as a continuous version of counterfactual probabilities [6] and allows to answer a retrospective question about the necessity of interventions: *what would have been the expected counterfactual outcome for some treatment, considering knowledge about both the factual treatment and the factual outcome?*

In our paper, we leverage geometric measure theory and differential topology to prove that, in general, the ignorance interval of ECOU [ECOT] has non-informative bounds. We show theoretically that this happens immediately when we relax the assumptions of (i) that the latent noise and the outcome have the same dimensionality and (ii) that the functions in the SCMs are monotonous (and assume they are continuously differentiable). As a remedy, we propose a novel ***Curvature Sensitivity Model*** (CSM), in which we bound the curvature of the level sets of the functions and thus yield informative bounds. We further show that we obtain the BGMs from [93] as a special case when setting the curvature to zero. Likewise, we yield non-informative bounds when setting it to infinity. Therefore, our CSM provides a sufficient condition for the partial counterfactual identification of the continuous outcomes with informative bounds.

We develop an instantiation of CSM in the form of a novel deep generative model, which we call *Augmented Pseudo-Invertible Decoder* (APID). Our APID uses (i) residual normalizing flows with (ii) variational augmentations to perform the task of partial counterfactual inference. Specifically, our APID allows us to (1) fit the observational/interventional data, (2) perform abduction-action-prediction in a differentiable fashion, and (3) bound the curvature of the SCM functions, thus yielding informative bounds for the whole ignorance interval. Finally, we demonstrate its effectiveness across several numerical experiments.

Overall, our **main contributions** are following:[1]

1. We prove that the expected counterfactual outcome of [un]treated has non-informative bounds in the class of continuously differentiable functions of SCMs.
2. We propose a novel *Curvature Sensitivity Model* (CSM) to obtain informative bounds. Our CSM is the first sensitivity model for the partial counterfactual identification of continuous outcomes in Markovian SCMs.
3. We introduce a novel deep generative model called *Augmented Pseudo-Invertible Decoder* (APID) to perform partial counterfactual inference under our CSM. We further validate it numerically.

## 2 Related Work

We briefly summarize prior works on (1) point and (2) partial counterfactual identification below but emphasize that none of them can be straightforwardly extended to our setting. We provide an extended literature overview in Appendix A.

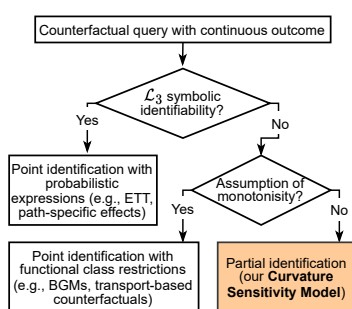

Figure 2: Flow chart of identifiability for a counterfactual query.

**(1) Point counterfactual identification** has been recently addressed through neural methods [21, 29, 31, 70, 84, 100, 113, 114, 117, 118, 134, 135, 140] but without identifiability results.

To ensure identifiability, prior works usually make use of (i) symbolic identifiability methods or (ii) put restrictive assumptions on the model class, if (i) led to non-identifiability (see Fig. 2). (i) Symbolic (non-parametric) identifiability methods [27, 121] aim to provide a symbolic probabilistic expression suitable for point identification if a counterfactual query can be expressed via lower-layer information only. Examples of the latter include the effect of treatment of the treated (ETT) [119] and path-specific effects [122, 146]. However, these are *not* suited for partial counterfactual identification.

Alternatively, identifiability can be achieved by (ii) making restrictive but unrealistic assumptions about the SCMs or the data generating mechanism [4, 32, 59, 66, 93, 96, 123, 148]. A notable example is the BGMs [93], which assumes a Markovian SCM where the functions in the SCMs must be monotonous (strictly increasing or decreasing) with respect to the latent noise. The latter assumption effectively sets the dimensionality of the latent noise to the same dimensionality as the observed (endogenous) variables, yet this is unrealistic in medicine. As a remedy, we depart from *point* identification and instead aim at *partial* counterfactual identification, which allows us to *relax* the assumptions of BGMs.

Kilbertus et al. [69] build sensitivity models for unobserved confounding in semi-Markovian SCMs, but still assume a restricted functional class of SCMs, namely, an additive noise model [66, 123]. We, on the other hand, build a sensitivity model around the extended class of functions in the SCMs, which is *non-trivial* even in the Markovian SCMs.

**(2) Partial counterfactual identification** has been studied previously, but only either for (i) *discrete* SCMs [97, 103, 129, 137, 141, 142, 143, 147], or for (ii) specific counterfactual queries with *informative bounds* [2, 4, 37]. Yet, these (i) do *not* generalize to the continuous setting; and (ii) are specifically tailored for certain queries with *informative bounds*, such as the variance of the treatment effects, and, thus, are not applicable to our (non-informative) ECOU [ECOT].

Likewise, it is also *not* possible to extend partial *interventional* identification of continuous outcomes [35, 49, 60, 61, 126] from the $\mathcal{L}_2$ to $\mathcal{L}_3$, unless it explicitly assumes an underlying SCM.

**Research gap.** To the best of our knowledge, we are the first to propose a sensitivity model for partial counterfactual identification of continuous outcomes in Markovian SCMs.

---

[1]Code is available at `https://github.com/Valentyn1997/CSM-APID`.

# 3 Partial Counterfactual Identification of Continuous Outcomes

In the following, we derive one of our main results: the ignorance interval of the ECOU [ECOT] has non-informative bounds if we relax the assumptions that (i) both the outcome and the latent noise are of the same dimensionality and that (ii) the functions in the SCMs are monotonous.

## 3.1 Preliminaries

**Notation.** Capital letters $A, Y, U$, denote random variables and small letters $a, y, u$ their realizations from corresponding domains $\mathcal{A}, \mathcal{Y}, \mathcal{U}$. Bold capital letters such as $\mathbf{U} = \{U_1, \ldots, U_n\}$ denote finite sets of random variables. Further, $\mathbb{P}(Y)$ is an (observational) distribution of $Y$; $\mathbb{P}(Y \mid a) = \mathbb{P}(Y \mid A = a)$ is a conditional (observational) distribution; $\mathbb{P}(Y_{A=a}) = \mathbb{P}(Y_a)$ an interventional distribution; and $\mathbb{P}(Y_{A=a} \mid A' = a', Y' = y') = \mathbb{P}(Y_a \mid a', y')$ a counterfactual distribution. We use a superscript such as in $\mathbb{P}^{\mathcal{M}}$ to indicate distributions that are induced by the SCM $\mathcal{M}$. We denote the conditional cumulative distribution function (CDF) of $\mathbb{P}(Y \mid a)$ by $\mathbb{F}_a(y)$. We use $\mathbb{P}(Y = \cdot)$ to denote a density or probability mass function of $Y$ and $\mathbb{E}(Y) = \int y \, d\mathbb{P}(y)$ to refer to its expected value. Interventional and counterfactual densities or probability mass functions are defined accordingly.

A function is said to be in class $C^k$, $k \geq 0$, if its $k$-th derivative exists and is continuous. Let $\|\cdot\|_2$ denote the $L_2$-norm, $\nabla_x f(x)$ a gradient of $f(x)$, and $\mathrm{Hess}_x f(x)$ a Hessian matrix of $f(x)$. The pushforward distribution or pushforward measure is defined as a transfer of a (probability) measure $\mathbb{P}$ with a measurable function $f$, which we denote as $f_\sharp \mathbb{P}$.

**SCMs.** We follow the standard notation of SCMs as in [8, 13, 101, 137]. An SCM $\mathcal{M}$ is defined as a tuple $\langle \mathbf{U}, \mathbf{V}, \mathbb{P}(\mathbf{U}), \mathcal{F} \rangle$ with latent (exogenous) noise variables $\mathbf{U}$, observed (endogenous) variables $\mathbf{V} = \{V_1, \ldots, V_n\}$, a distribution of latent noise variables $\mathbb{P}(\mathbf{U})$, and a collection of functions $\mathcal{F} = \{f_{V_1}, \ldots, f_{V_n}\}$. Each function is a measurable map from the corresponding domains of $\mathbf{U}_{V_i} \cup \mathbf{Pa}_{V_i}$ to $V_i$, where $\mathbf{U}_{V_i} \subseteq \mathbf{U}$ and $\mathbf{Pa}_{V_i} \subseteq \mathbf{V} \setminus V_i$ are parents of the observed variable $V_i$. Therefore, the functions $\mathcal{F}$ induce a pushforward distribution of observed variables, i.e., $\mathbb{P}(\mathbf{V}) = \mathcal{F}_\sharp \mathbb{P}(\mathbf{U})$. Each $V_i$ is thus deterministic (non-random), conditionally on its parents, i.e., $v_i \leftarrow f_{V_i}(\mathbf{pa}_{V_i}, \mathbf{u}_{V_i})$. Each SCM $\mathcal{M}$ induces an (augmented) causal diagram $\mathcal{G}(\mathcal{M})$, which is assumed to be acyclic.

We provide a background on geometric measure theory and differential geometry in Appendix B.

## 3.2 Counterfactual Non-Identifiability

In the following, we relax the main assumption of bijective generation mechanisms (BGMs) [93], i.e., that all the functions in Markovian SCMs are monotonous (strictly increasing) with respect to the latent noise variable. To this end, we let the latent noise variables have arbitrary dimensionality and further consider functions to be of class $C^k$. We then show that counterfactual distributions under this relaxation are non-identifiable from $\mathcal{L}_1$ or $\mathcal{L}_2$ data.

**Definition 1** (Bivariate Markovian SCMs of class $C^k$ and $d$-dimension latent noise). *Let $\mathfrak{B}(C^k, d)$ denote the class of SCMs $\mathcal{M} = \langle \mathbf{U}, \mathbf{V}, \mathbb{P}(\mathbf{U}), \mathcal{F} \rangle$ with the following endogenous and latent noise variables: $\mathbf{U} = \{U_A \in \{0, 1\}, U_Y \in [0, 1]^d\}$ and $\mathbf{V} = \{A \in \{0, 1\}, Y \in \mathbb{R}\}$, for $d \geq 1$. The latent noise variables have the following distributions $\mathbb{P}(\mathbf{U})$: $U_A \sim \mathrm{Bern}(p_A)$, $0 < p_A < 1$, and $U_Y \sim \mathrm{Unif}(0, 1)^d$ and are all mutually independent [2]. The functions are $\mathcal{F} = \{f_A(U_A), f_Y(A, U_Y)\}$ and $f_Y(a, \cdot) \in C^k$ for $u_Y \in (0, 1)^d$ $\forall a \in \{0, 1\}$.*

All SCMs in $\mathfrak{B}(C^k, d)$ induce similar causal diagrams $\mathcal{G}(\mathcal{M})$, where only the number of latent noise variables $d$ differs. Further, it follows from the above definition that BGMs are a special case of $\mathfrak{B}(C^1, 1)$, where the functions $f_Y(a, \cdot)$ are monotonous (strictly increasing).

**Task: counterfactual inference.** Counterfactual queries are at layer $\mathcal{L}_3$ of the causality ladder and are defined as probabilistic expressions with random variables belonging to several worlds, which are in logical contradiction with each other [8, 27]. For instance, $\mathbb{P}^{\mathcal{M}}(Y_a = y, Y_{a'} = y), a \neq a'$ for an SCM $\mathcal{M}$ from $\mathfrak{B}(C^k, d)$. In this paper, we focus on the *expected counterfactual outcome of [un]treated* ECOU [ECOT], which we denote by $Q^{\mathcal{M}}_{a' \to a}(y') = \mathbb{E}^{\mathcal{M}}(Y_a \mid a', y')$.

---

[2]Uniformity of the latent noise does not restrict the definition, see the discussion in Appendix D.

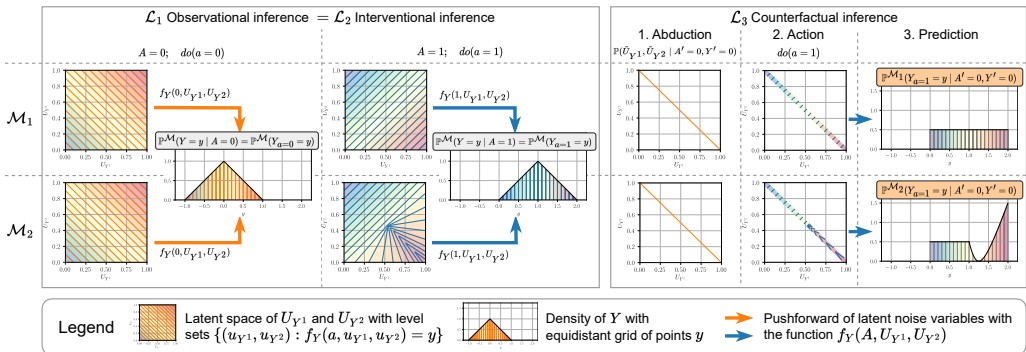

Figure 3: Inference of observational and interventional distributions (left) and counterfactual distributions (right) for SCMs $\mathcal{M}_1$ and $\mathcal{M}_2$ from Example 1. *Left:* the observational query $\mathbb{P}^{\mathcal{M}}(Y = y \mid a)$ coincides with the interventional query $\mathbb{P}^{\mathcal{M}}(Y_a = y)$ for each $\mathcal{M}_1$ and $\mathcal{M}_2$. *Right:* the counterfactual queries can still differ substantially for $\mathcal{M}_1$ and $\mathcal{M}_2$, thus giving a vastly different counterfactual outcome distribution of the untreated, $\mathbb{P}^{\mathcal{M}}(Y_1 \mid A' = 0, Y' = 0)$. Thus, $\mathcal{L}_3$ queries are non-identifiable from $\mathcal{L}_1$ or $\mathcal{L}_2$ information.

For an SCM $\mathcal{M}$ from $\mathfrak{B}(C^k, d)$, ECOU [ECOT] can be inferred by the following three steps. (1) The *abduction* step infers a posterior distribution of the latent noise variables, conditioned on the evidence, i.e., $\mathbb{P}(U_Y \mid a', y')$. This posterior distribution is defined on the level set of the factual function given by $\{u_Y : f_Y(a', u_Y) = y'\}$, i.e., all points in the latent noise space mapped to $y'$. (2) The *action* step alters the function for $Y$ to $f_Y(a, U_Y)$. (3) The *prediction* step is done by a pushforward of the posterior distribution with the altered function, i.e. $f_Y(a, \cdot)_\sharp \mathbb{P}(U_Y \mid a', y')$. Afterwards, the expectation of it is then evaluated.

The existence of counterfactual queries, which are non-identifiable with $\mathcal{L}_1$ or $\mathcal{L}_2$ data in Markovian SCMs, was previously shown in [137] (Ex. 8, App. D3) for discrete outcomes and in [13, 30] (Ex. D.7) for continuous outcomes. Here, we construct an important example to (1) show non-identifiability of counterfactual queries from $\mathcal{L}_1$ or $\mathcal{L}_2$ data under our relaxation from Def. 1, and to consequently (2) give some intuition on informativity of the bounds of the ignorance interval, which we will formalize later.

**Example 1** (Counterfactual non-identifiability in Markovian SCMs)**.** *Let $\mathcal{M}_1$ and $\mathcal{M}_2$ be two Markovian SCMs from $\mathfrak{B}(C^0, 2)$ with the following functions for $Y$:*

$$\mathcal{M}_1 : f_Y(A, U_{Y^1}, U_{Y^2}) = A\left(U_{Y^1} - U_{Y^2} + 1\right) + (1 - A)\left(U_{Y^1} + U_{Y^2} - 1\right)\},$$

$$\mathcal{M}_2 : f_Y(A, U_{Y^1}, U_{Y^2}) = \begin{cases} U_{Y^1} + U_{Y^2} - 1, & A = 0, \\ U_{Y^1} - U_{Y^2} + 1, & A = 1 \wedge (0 \le U_{Y^1} \le 1) \wedge (U_{Y^1} \le U_{Y^2} \le 1), \\ F^{-1}(0, U_{Y^1}, U_{Y^2}), & \text{otherwise}, \end{cases}$$

*where $F^{-1}(0, U_{Y^1}, U_{Y^2})$ is the solution in $Y$ of the implicitly defined function $F(Y, U_{Y^1}, U_{Y^2}) = U_{Y^1} - U_{Y^2} - 2(Y - 1)\left|-U_{Y^1} - U_{Y^2} + 1\right| - 1 + \sqrt{(Y - 2)^2\left(8(Y - 1)^2 + 1\right)} = 0$.*

*It turns out that the SCMs $\mathcal{M}_1$ and $\mathcal{M}_2$ are observationally and interventionally equivalent (relative to the outcome $Y$). That is, they induce the same set of $\mathcal{L}_1$ and $\mathcal{L}_2$ queries. For both SCMs, it can be easily inferred that a pushforward of uniform latent noise variables $U_{Y^1}$ and $U_{Y^2}$ with $f_Y(A, U_{Y^1}, U_{Y^2})$ is a symmetric triangular distribution with support $\mathcal{Y}_0 = [-1, 1]$ for $A = 0$ and $\mathcal{Y}_1 = [0, 2]$ for $A = 1$, respectively. We plot the level sets of $f_Y(A, U_{Y^1}, U_{Y^2})$ for both SCMs in Fig. 3 (left). The pushforward of the latent noise variables preserves the transported mass; i. e., note the equality of (1) the area of each colored band between the two level sets in the latent noise space and (2) the area of the corresponding band under the density graph of $Y$ Fig. 3 (left).*

*Despite the equivalence of $\mathcal{L}_1$ and $\mathcal{L}_2$, the SCMs differ in their counterfactuals; see Fig. 3 (right). For example, the counterfactual outcome distribution of untreated, $\mathbb{P}^{\mathcal{M}}(Y_1 = y \mid A' = 0, Y' = 0)$, has different densities for both SCMs $\mathcal{M}_1$ and $\mathcal{M}_2$. Further, the ECOU, $Q_{0 \to 1}^{\mathcal{M}}(0) = \mathbb{E}^{\mathcal{M}}(Y_1 \mid A' = 0, Y' = 0)$, is different for both SCMs, i. e., $Q_{0 \to 1}^{\mathcal{M}_1}(0) = 1$ and $Q_{0 \to 1}^{\mathcal{M}_2} \approx 1.114$. Further details for the example are in Appendix C.*

The example provides an intuition that motivates how we generate informative bounds later. By "bending" the bundle of counterfactual level sets (in blue in Fig. 3 left) around the factual level set

(in orange in Fig. 3, right), we can transform more and more mass to the bound of the support. We later extend this idea to the ignorance interval of the ECOU. Importantly, after "bending" the bundle of level sets, we still must make sure that the original observational/interventional distribution is preserved.

## 3.3 Partial Counterfactual Identification and Non-Informative Bounds

We now formulate the task of partial counterfactual identification. To do so, we first present two lemmas that show how we can infer the densities of observational and counterfactual distributions from both the latent noise distributions and $C^1$ functions in SCMs of class $\mathfrak{B}(C^1, d)$.

**Lemma 1** (Observational distribution as a pushforward with $f_Y$). *Let $\mathcal{M} \in \mathfrak{B}(C^1, d)$. Then, the density of the observational distribution, induced by $\mathcal{M}$, is*

$$\mathbb{P}^{\mathcal{M}}(Y = y \mid a) = \int_{E(y,a)} \frac{1}{\|\nabla_{u_Y} f_Y(a, u_Y)\|_2} \, \mathrm{d}\mathcal{H}^{d-1}(u_Y), \tag{1}$$

*where $E(y, a)$ is a level set (preimage) of $y$, i.e., $E(y, a) = \{u_Y \in [0,1]^d : f_Y(a, u_Y) = y\}$, and $\mathcal{H}^{d-1}(u_Y)$ is the Hausdorff measure (see Appendix B for the definition).*

We provide an example in Appendix C where we show the application of Lemma 1 (therein, we derive the standard normal distribution as a pushforward using the Box-Müller transformation). Lemma 1 is a generalization of the well-known change of variables formula. This is easy to see, when we set $d = 1$, so that $\mathbb{P}^{\mathcal{M}}(Y = y \mid a) = \sum_{u_Y \in E(y,a)} |\nabla_{u_Y} f_Y(a, u_Y)|^{-1}$. Furthermore, the function $f_Y(a, u_Y)$ can be restored (up to a sign) from the observational distribution, if it is monotonous in $u_Y$, such as in BGMs [93]. In this case, the function coincides (up to a sign) with the inverse CDF of the observed distribution, i.e., $f_Y(a, u_Y) = \mathbb{F}_a^{-1}(\pm u_Y \mp 0.5 + 0.5)$ (see Corollary 1 in Appendix D).

**Lemma 2.** *Let $\mathcal{M} \in \mathfrak{B}(C^1, d)$. Then, the density of the counterfactual outcome distribution of the [un]treated is*

$$\mathbb{P}^{\mathcal{M}}(Y_a = y \mid a', y') = \frac{1}{\mathbb{P}^{\mathcal{M}}(Y = y' \mid a')} \int_{E(y',a')} \frac{\delta(f_Y(a, u_Y) - y)}{\|\nabla_{u_Y} f_Y(a', u_Y)\|_2} \, \mathrm{d}\mathcal{H}^{d-1}(u_Y), \tag{2}$$

*where $\delta(\cdot)$ is a Dirac delta function, and the expected counterfactual outcome of the [un]treated, i.e., ECOU [ECOT], is*

$$Q_{a' \to a}^{\mathcal{M}}(y') = \mathbb{E}^{\mathcal{M}}(Y_a \mid a', y') = \frac{1}{\mathbb{P}^{\mathcal{M}}(Y = y' \mid a')} \int_{E(y',a')} \frac{f_Y(a, u_Y)}{\|\nabla_{u_Y} f_Y(a', u_Y)\|_2} \, \mathrm{d}\mathcal{H}^{d-1}(u_Y), \tag{3}$$

*where $E(y', a')$ is a (factual) level set of $y'$, i.e., $E(y', a') = \{u_Y \in [0,1]^d : f_Y(a', u_Y) = y'\}$ and $a' \neq a$.*

Equations (2) and (3) implicitly combine all three steps of the abduction-action-prediction procedure: (1) *abduction* infers the level sets $E(y', a')$ with the corresponding Hausdorff measure $\mathcal{H}^{d-1}(u_Y)$; (2) *action* uses the counterfactual function $f_Y(a, u_Y)$; and (3) *prediction* evaluates the overall integral. In the specific case of $d = 1$ and a monotonous function $f_Y(a, u_Y)$ with respect to $u_Y$, we obtain two deterministic counterfactuals, which are identifiable from observational distribution, i.e., $Q_{a' \to a}^{\mathcal{M}}(y') = \mathbb{F}_a^{-1}(\pm \mathbb{F}_{a'}(y') \mp 0.5 + 0.5)$. For details, see Corollary 3 in Appendix D. For larger $d$, as already shown in Example 1, both the density and ECOU [ECOT] can take arbitrary values for the same observational (or interventional) distribution.

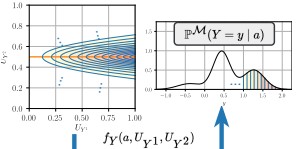

Figure 4: "Bending" the bundle of counterfactual level sets $\{E(y, a) : y \in [1, 2]\}$ in blue around the factual level set $E(y', a')$ in orange.

**Definition 2** (Partial identification of ECOU (ECOT) in class $\mathfrak{B}(C^k, d), k \geq 1$). *Given the continuous observational distribution $\mathbb{P}(Y \mid a)$ for some SCM of class $\mathfrak{B}(C^k, d)$. Then, partial counterfactual identification aims to find bounds of the ignorance interval $[\underline{Q_{a' \to a}(y')}, \overline{Q_{a' \to a}(y')}]$ given by*

$$\underline{Q_{a' \to a}(y')} = \inf_{\mathcal{M} \in \mathfrak{B}(C^k, d)} Q_{a' \to a}^{\mathcal{M}}(y') \quad s.t. \ \forall a \in \{0, 1\} : \mathbb{P}(Y \mid a) = \mathbb{P}^{\mathcal{M}}(Y \mid a), \tag{4}$$

$$\overline{Q_{a' \to a}(y')} = \sup_{\mathcal{M} \in \mathfrak{B}(C^k, d)} Q_{a' \to a}^{\mathcal{M}}(y') \quad s.t. \ \forall a \in \{0, 1\} : \mathbb{P}(Y \mid a) = \mathbb{P}^{\mathcal{M}}(Y \mid a), \tag{5}$$

*where $\mathbb{P}^{\mathcal{M}}(Y \mid a)$ is given by Eq. (1) and $Q_{a' \to a}^{\mathcal{M}}(y')$ by Eq. (3).*

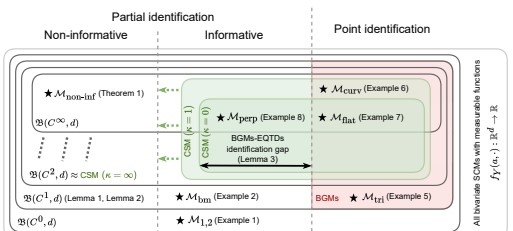

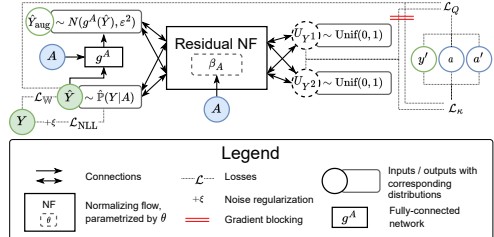

Figure 5: Venn diagram of different SCM classes $\mathfrak{B}(C^k, d)$ arranged by partial vs. point point identification. See our CSM with different $\kappa$. We further show the relationships between different classes of SCMs $\mathfrak{B}(C^k, d)$ and statements about them, e.g., Lemmas 1 and 2. The referenced examples are in the Appendix C.

Figure 6: Overview of our *Augmented Pseudo-Invertible Decoder* (APID). Our APID uses (i) two residual normalizing flows for each treatment $a \in \{0, 1\}$, respectively; and (ii) variational augmentations, $\hat{Y}_{\text{aug}}$. Together, both components enable the implementation of CSM under Assumption $\kappa$.

Hence, the task of partial counterfactual identification in class $\mathfrak{B}(C^k, d)$ can be reduced to a constrained variational problem, namely, a constrained optimization of ECOU [ECOT] with respect to $f_Y(a, \cdot) \in C^k$. Using Lemma 2, we see that, e.g., ECOU [ECOT] can be made arbitrarily large (or small) for the same observational distribution in two ways. First, this can be done by changing the factual function, $f_Y(a', \cdot)$, i.e., if we increase the proportion of the volume of the factual level set $E(y', a')$ that intersects only a certain bundle of counterfactual level sets. Second, this can be done by modifying the counterfactual function, $f_Y(a, \cdot)$, by "bending" the bundle of counterfactual level sets around the factual level set. The latter is schematically shown in Fig. 4. We formalize this important observation in the following theorem.

**Theorem 1** (Non-informative bounds of ECOU (ECOT)). *Let the continuous observational distribution $\mathbb{P}(Y \mid a)$ be induced by some SCM of class $\mathfrak{B}(C^\infty, d)$. Let $\mathbb{P}(Y \mid a)$ have a compact support $\mathcal{Y}_a = [l_a, u_a]$ and be of finite density $\mathbb{P}(Y = y \mid a) < +\infty$. Then, the ignorance interval for the partial identification of the ECOU [ECOT] of class $\mathfrak{B}(C^\infty, d)$, $d \geq 2$, has non-informative bounds: $\underline{Q_{a' \to a}(y')} = l_a$ and $\overline{Q_{a' \to a}(y')} = u_a$.*

Theorem 1 implies that, no matter how smooth the class of functions is, the partial identification of ECOU [ECOT] will have non-informative bounds. Hence, with the current set of assumptions, there is no utility in considering more general classes. This includes various functions, such as $C^0$ and the class of all measurable functions $f_Y(a, \cdot) : \mathbb{R}^d \to \mathbb{R}$, as the latter includes $C^\infty$ functions. In the following, we introduce a sensitivity model which nevertheless allows us to obtain informative bounds.

## 4 Curvature Sensitivity Model

In the following, we develop our *Curvature Sensitivity Model* (CSM) to restrict the class $\mathfrak{B}(C^k, d)$ so that the ECOU [ECOT] obtains informative bounds. Our CSM uses the intuition from the proof of Theorem 1 in that, to construct the non-informative SCM $\mathcal{M}_{\text{non-inf}}$, we have to "bend" the bundle of the counterfactual level sets. As a result, our CSM provides sufficient conditions for informative bounds in the class $\mathfrak{B}(C^2, d)$ by bounding the principal curvatures of level sets globally.

**Assumption $\kappa$.** *Let $\mathcal{M}$ be of class $\mathfrak{B}(C^2, d), d \geq 2$. Let $E(y, a)$ be the level sets of functions $f_Y(a, u_Y)$ for $a \in \{0, 1\}$, which are thus $d - 1$-dimensional smooth manifolds. Let us assume that principal curvatures $i \in \{1, \ldots, d-1\}$ exist at every point $u_Y \in E(y, a)$ for every $a \in \{0, 1\}$ and $y \in (l_a, u_a) \subset \mathcal{Y}_a$, and let us denote them as $\kappa_i(u_Y)$. Then, we assume that $\kappa \geq 0$ is the upper bound of the maximal absolute principal curvature for every $y$, $a$, and $u_Y \in E(y, a)$:*

$$\kappa = \max_{a \in \{0,1\}, y \in (l_a, u_a), u_Y \in E(y,a)} \quad \max_{i \in \{1, \ldots, d-1\}} |\kappa_i(u_Y)|. \tag{6}$$

Principal curvatures can be thought of as a measure of the non-linearity of the level sets, and, when they are all close to zero at some point, the level set manifold can be locally approximated by a flat hyperplane. In brevity, principal curvatures can be defined via the first- and second-order partial derivatives of $f_Y(a, \cdot)$, so that they describe the degrees of curvature of a manifold in different directions. We refer to Appendix B for a formal definition of the principal curvatures $\kappa_i$. An example is in Appendix C.

Now, we state the main result of our paper that our CSM allows us to obtain informative bounds for ECOU [ECOT].

**Theorem 2** (Informative bounds with our CSM). *Let the continuous observational distribution* $\mathbb{P}(Y \mid a)$ *be induced by some SCM of class* $\mathfrak{B}(C^2, d), d \geq 2$, *which satisfies Assumption* $\kappa$. *Let* $\mathbb{P}(Y \mid a)$ *have a compact support* $\mathcal{Y}_a = [l_a, u_a]$. *Then, the ignorance interval for the partial identification of ECOU [ECOT] of class* $\mathfrak{B}(C^2, d)$ *has informative bounds, dependent on* $\kappa$ *and* $d$, *which are given by* $\underline{Q_{a' \to a}(y')} = l(\kappa, d) > l_a$ *and* $\overline{Q_{a' \to a}(y')} = u(\kappa, d) < u_a$.

Theorem 2 has several important implications. (1) Our CSM is applicable to a wide class of functions $\mathfrak{B}(C^2, d)$, for which the principal curvature is well defined. (2) We show the relationships between different classes $\mathfrak{B}(C^k, d)$ and our CSM ($\kappa$) in terms of identifiability in Fig. 5. For example, by increasing $\kappa$, we cover a larger class of functions, and the bounds on ECOU [ECOT] expand (see Corollary 4 in Appendix D). For infinite curvature, our CSM almost coincides with the entire $\mathfrak{B}(C^2, d)$. (3) Our CSM ($\kappa$) with $\kappa \geq 0$ always contains both (i) identifiable SCMs and (ii) (informative) non-identifiable SCMs. Examples of (i) include SCMs for which the bundles of the level sets coincide for both treatments, i. e., $\{E(y, a) : y \in \mathcal{Y}_a\} = \{E(y, a') : y \in \mathcal{Y}_{a'}\}$. In this case, it is always possible to find an equivalent BGM when the level sets are flat and thus $\kappa = 0$ (see $\mathcal{M}_{\text{flat}}$ from Example 6 in the Appendix C) or curved and thus $\kappa = 1$ (see $\mathcal{M}_{\text{curv}}$ from Example 7 in the Appendix C). For (ii), we see that, even when we set $\kappa = 0$, we can obtain non-identifiability with informative bounds. An example is when we align the bundle of level sets perpendicularly to each other for both treatments, as in $\mathcal{M}_{\text{perp}}$ from Example 8 in the Appendix C. We formalize this observation with the following Lemma.

**Lemma 3** (BGMs-EQTDs identification gap of CSM($\kappa = 0$)). *Let the assumptions of Theorem 2 hold and let* $\kappa = 0$. *Then the ignorance intervals for the partial identification of ECOU [ECOT] of class* $\mathfrak{B}(C^2, d)$ *are defined by min/max between BGMs bounds and expectations of quantile-truncated distributions (EQTDs):*

$$\underline{Q_{a' \to a}(y')} / \overline{Q_{a' \to a}(y')} = \min / \max \{BGM_+(y'), BGM_-(y'), EQTD_l(y'), EQTD_u(y')\} \quad (7)$$

$$BGM_+(y') = \mathbb{F}_a^{-1}(\mathbb{F}_{a'}(y')) \qquad BGM_-(y') = \mathbb{F}_a^{-1}(1 - \mathbb{F}_{a'}(y')), \quad (8)$$

$$EQTD_l(y') = \mathbb{E}\left(Y \mid a, Y < \mathbb{F}_a^{-1}(1 - 2\,|0.5 - \mathbb{F}_{a'}(y')|)\right), \quad (9)$$

$$EQTD_u(y') = \mathbb{E}\left(Y \mid a, Y > \mathbb{F}_a^{-1}(2\,|\mathbb{F}_{a'}(y') - 0.5|)\right), \quad (10)$$

*where* $\mathbb{E}(Y \mid Y < \cdot), \mathbb{E}(Y \mid Y > \cdot)$ *are expectations of truncated distributions.*

Lemma 3 shows how close we can get to the point identification after setting $\kappa = 0$ in our CSM. In particular, with CSM ($\kappa = 0$) the ignorance interval still contains BGMs bounds and EQTDs, i. e., this is the identification gap of CSM.[3] Hence, to obtain full point identification with our CSM, additional assumptions or constraints are required. Notably, we can not assume monotonicity, as there is no conventional notion of monotonicity for functions from $\mathbb{R}^d$ to $\mathbb{R}$.

We make another observation regarding the choice of the latent noise dimensionality $d$. In practice, it is sufficient to choose $d = 2$ (without further assumptions on the latent noise space). This choice is practical as we only have to enforce a single principal curvature, which reduces the computational burden, i.e.,

$$\kappa_1(u_Y) = -\frac{1}{2}\nabla_{u_Y}\left(\frac{\nabla_{u_Y} f_Y(a, u_Y)}{\|\nabla_{u_Y} f_Y(a, u_Y)\|_2}\right), \quad u_Y \in E(y, a). \quad (11)$$

Importantly, we do not lose generality with $d = 2$, as we still cover the entire identifiability spectrum by varying $\kappa$ (see Corollary 5 in Appendix D). We discuss potential extensions of our CSM in Appendix E.

**Interpretation of** $\kappa$. The sensitivity parameter $\kappa$ lends to a natural interpretation. Specifically, it can be interpreted as *a level of non-linearity between the outcome and its latent noises that interact with the treatment*. For example, when (i) the treatment does not interact with any of the latent noise variables, then we can assume w.l.o.g. a BGM, which, in turn, yields deterministic counterfactual outcomes. There is no loss of generality, as all the other SCMs with $d > 1$ are equivalent to this

---

[3]In general, BGMs bounds and EQTDs have different values, especially for skewed distributions. Nevertheless, it can be shown, that they approximately coincide as $y'$ goes to $l_{a'}$ or $u_{a'}$.

BGM (see Examples 6 and 7 in Appendix C). On the other hand, (ii) when the treatment interacts with some latent noise variables, then the counterfactual outcomes become random, and we cannot assume a BGM but we have to use our CSM. In this case, $d$ corresponds to the number of latent noise variables which interact with the treatment. Hence, $\kappa$ bounds the level of non-linearity between the outcome and the noise variables. More formally, $\kappa$, as a principal curvature, can be seen as the largest coefficient of the second-order term in the Taylor expansion of the level set (see Eq. (19) in Appendix B). This interpretation of the $\kappa$ goes along with human intuition [20]: when we try to imagine counterfactual outcomes, we tend to "traverse" all the possible scenarios which could lead to a certain value. If we allow for highly non-linear scenarios, which interact with treatment, we also allow for more extreme counterfactuals, e. g., interactions between treatment and rare genetic conditions.

## 5 Augmented Pseudo-Invertible Decoder

We now introduce an instantiation of our CSM: a novel deep generative model called *Augmented Pseudo-Invertible Decoder* (APID) to perform partial counterfactual identification under our CSM ($\kappa$) of class $\mathfrak{B}(C^2, 2)$.

**Architecture:** The two main components of our APID are (1) residual normalizing flows with (2) variational augmentations (see Fig. 6). The first component are two-dimensional normalizing flows [107, 125] to estimate the function $f_Y(a, u_Y), u_Y \in [0, 1]^2$, separately for each treatment $a \in \{0, 1\}$. Specifically, we use residual normalizing flows [24] due to their ability to model free-form Jacobians (see the extended discussion about the choice of the normalizing flow in Appendix F). However, two-dimensional normalizing flows can only model invertible transformations, while the function $\hat{f}_Y(a, \cdot) : [0, 1]^2 \to \mathcal{Y}_a \subset \mathbb{R}$ is non-invertible. To address this, we employ an approach of pseudo-invertible flows [10, 54], namely, the variational augmentations [22], as described in the following. The second component in our APID are variational augmentations [22]. Here, we augment the estimated outcome variable $\hat{Y}$ with the variationally sampled $\hat{Y}_{\text{aug}} \sim N(g^a(\hat{Y}), \varepsilon^2)$, where $g^a(\cdot)$ is a fully-connected neural network, and $\varepsilon^2 > 0$ is a hyperparameter. Using the variational augmentations, our APID then models $\hat{f}_Y(a, \cdot)$ through a two-dimensional transformation $\hat{F}_a = (\hat{f}_{Y_{\text{aug}}}(a, \cdot), \hat{f}_Y(a, \cdot)) : [0, 1]^2 \to \mathbb{R} \times \mathcal{Y}_a$. We refer to the Appendix F for further details.

**Inference:** Our APID proceeds in first steps: (P1) it first fits the observational/interventional distribution $\mathbb{P}(Y \mid a)$, given the observed samples; (P2) it then performs counterfactual inference of ECOU [ECOT] in a differentiable fashion, which, therefore, can be maximized/minimized jointly with other objectives; and (P3) it finally penalize functions with large principal curvatures of the level sets, by using automatic differentiation of the estimated functions of the SCMs.

We achieve (P1)–(P3) through the help of variational augmentations: ● In (P1), we fit two two-dimensional normalizing flows with a negative log-likelihood (for the task of maximizing or minimizing ECOU [ECOT], respectively). ● In (P2), for each normalizing flow, we sample points from the level sets of $\hat{f}_Y(a, \cdot)$. The latter is crucial as it follows an abduction-action-prediction procedure and thus generates estimates of the ECOU [ECOT] in a differential fashion. To evaluate ECOU [ECOT], we first perform the *abduction* step with the inverse transformation of the factual normalizing flow, i.e., $\hat{F}_{a'}^{-1}$. This transformation maps the variationally augmented evidence, i. e., $(y', \{Y'_{\text{aug}}\}_{j=1}^b \sim N(g^a(y'), \varepsilon^2))$, where $b$ is the number of augmentations, to the latent noise space. Then, the *action* step selects the counterfactual normalizing flow via the transformation $\hat{F}_a$. Finally, the *prediction* step performs a pushforward of the latent noise space, which was inferred during the abduction step. Technical details are in Appendix F. ● In (P3), we enforce the curvature constraint of our CSM. Here, we use automatic differentiation as provided by deep learning libraries, which allows us to directly evaluate $\kappa_1(u_Y)$ according to Eq. (11).

**Training:** Our APID is trained based on observational data $\mathcal{D} = \{A_i, Y_i\}_{i=1}^n$ drawn i.i.d. from some SCM of class $\mathfrak{B}(C^2, 2)$. To fit our APID, we combine several losses in one optimization objective: (1) a negative log-likelihood loss $\mathcal{L}_{\text{NLL}}$ with noise regularization [110], which aims to fit the data distribution; (2) a counterfactual query loss $\mathcal{L}_Q$ with a coefficient $\lambda_Q$, which aims to maximize/minimize the ECOU [ECOT]; and (3) a curvature loss $\mathcal{L}_\kappa$ with coefficient $\lambda_\kappa$, which penalizes the curvature of the level sets. Both coefficients are hyperparameters. We provide details about the losses, the training algorithm, and hyperparameters in Appendix G. Importantly, we

incorporated several improvements to stabilize and speed up the training. For example, in addition to the negative log-likelihood, we added the Wasserstein loss $\mathcal{L}_{\mathbb{W}}$ to prevent posterior collapse [26, 132]. To speed up the training, we enforce the curvature constraint only on the counterfactual function, $\hat{f}_Y(a, \cdot)$, and only for the level set, which corresponds to the evaluated ECOU [ECOT], $u_Y \in E(\hat{Q}_{a' \to a}(y'), a)$.

## 6 Experiments

**Datasets.** To show the effectiveness of our APID at partial counterfactual identification, we use two synthetic datasets. This enables us to access the ground truth CDFs, quantile functions, and sampling. Then, with the help of Lemma 3, we can then compare our **APID** with the **BGMs-EQTDs identification gap** (= a special case of CSM with $\kappa = 0$). Both synthetic datasets comprise samples from observational distributions $\mathbb{P}(Y \mid a)$, which we assume to be induced by some (unknown) SCM of class $\mathfrak{B}(C^2, 2)$. In the first dataset, $\mathbb{P}(Y \mid 0) = \mathbb{P}(Y \mid 1)$ is the standard normal distribution, and in second, $\mathbb{P}(Y \mid 0)$ and $\mathbb{P}(Y \mid 1)$ are different mixtures of normal distributions. We draw $n_a = 1,000$ observations from $\mathbb{P}(Y \mid a)$ for each treatment $a \in \{0, 1\}$, so that $n = n_0 + n_1 = 2,000$. Although both distributions have infinite support, we consider the finite sample minimum and maximum as estimates of the support bounds $[\hat{l}_1, \hat{u}_1]$. Further details on our synthetic datasets are in Appendix H. In sum, the estimated bounds from our APID are consistent with the theoretical values, thus showing the effectiveness of our method.

**Results.** Fig. 7 shows the results of point/partial counterfactual identification of the ECOU, i.e., $\hat{Q}_{0 \to 1}(y')$, for different values of $y'$. Point identification with BGMs yields two curves, corresponding to strictly increasing and strictly decreasing functions. For partial identification with our APID, we set $\lambda_Q = 2.0$ and vary $\lambda_\kappa \in \{0.5, 1.0, 5.0, 10.0\}$ (higher values correspond to a higher curvature penalization). We observe that, as we increase $\lambda_\kappa$, the bounds are moving closer to the BGMs, and, as we decrease it, the bounds are becoming non-informative, namely, getting closer to $[\hat{l}_1, \hat{u}_1]$. We report additional results in Appendix H (e. g., for APID with $\lambda_Q = 1.0$).

**Case study with real-world data.** In Appendix I, we provide a real-world case study. Therein, we adopt our CSM to answer "what if" questions related to whether the lockdown was effective during the COVID-19 pandemic.

Figure 7: Results for partial counterfactual identification of the ECOU across two datasets. Reported: BGMs-EQTDs identification gap and mean bounds of APID over five runs.

## 7 Discussion

**Limitations.** Our CSM and its deep-learning implementation with APID have several limitations, which could be addressed in future work. First, given the value of $\kappa$, it is computationally infeasible to derive ground-truth bounds, even when the ground-truth observation density is known. The inference of these bounds would require solving a constrained optimization task including partial derivatives and Hausdorff integration. This is intractable, even for such simple distributions as standard normal. Second, the exact relationship between $\kappa$ and corresponding $\lambda_\kappa$ is unknown in APID, but they are known to be inversely related. Third, our APID sometimes suffers from computational instability, e. g., the bounds for the multi-modal distribution (see Fig. 7), are inaccurate, i. e., too tight. This happens as the gradual "bending" of the level sets is sometimes numerically unstable during training, and some runs omit "bending" at all as it would require passing through the high loss value region.

**Conclusion.** Our work is the first to present a sensitivity model for partial counterfactual identification of continuous outcomes in Markovian SCMs. Our work rests on the assumption of the bounded curvature of the level sets, yet which should be sufficiently broad and realistic to cover many models from physics and medicine. As a broader impact, we expect our bounds to be highly relevant for decision-making in safety-critical settings.

## Acknowledgments

SF acknowledges funding from the Swiss National Science Foundation (SNSF) via Grant 186932.

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

# A  Extended Related Work

## A.1  Counterfactual inference

In the following, we explain why existing work does not straightforwardly generalize to the partial counterfactual identification of continuous outcomes. In particular, we do the following:

1. We pinpoint that symbolic (non-parametric) counterfactual identifiability methods only provide the probabilistic expression suitable for point identification, if a certain query can be expressed via lower layer information.
2. We note that existing point identification methods do *not* have any guidance towards partial identification, and often do not even provide identifiability guarantees. In this case, valid point identification can only be achieved via additional assumptions either (i) about the SCM or (ii) about the data generating mechanism.
3. We discuss why methods for partial identification of (i) discrete counterfactual queries, (ii) continuous counterfactual queries with informative bounds, and (iii) continuous interventional queries can *not* be extended to our setting.

Ultimately, we summarize related works on both point and partial identification for two layers of causality, namely interventional and counterfactual, in Table 1.

Table 1: Overview of methods for point and partial identification, assuming a known causal diagram. Relevant methods are highlighted in yellow.

| Layer | M/SM | Symbolic identifiability | Point identification methods | Partial identification methods | | |
|---|---|---|---|---|---|---|
| | | | | Discrete outcomes | Continuous outcomes | |
| | | | | | Informative bounds | Non-informative bounds |
| $\mathcal{L}_2$ Interventional | M | Always via backdoor criterion [8] | Deep generative models [74, 144] | — | — | — |
| | SM | Do-calculus & rules of probability [56, 79, 120] | Potential outcomes framework [14, 28, 52, 90, 111, 130] ; binary IV [39, 58, 131] ; proxy variables [85, 92] | Partially observed back-/front-door variables [81]; canonical SCM [136] | No-assumptions bound [88]; MSM [15, 35, 40, 41, 60, 61, 62, 98, 126]; outcome sensitivity models [15, 106]; confounding functions [11, 17, 109]; noisy proxy variables [49]; IV [48, 55, 68, 145]; clustered DAGs [99] | MSM [40, 41, 89]; ATD [5] |
| $\mathcal{L}_3$ Counterfactual | M | Parallel worlds networks [3, 121], counterfactual unnesting theorem [27] | Deep generative models [21, 29, 70, 100, 113, 114, 117, 118] ; Markovian BGMs [59, 66, 93, 94, 123, 148] ; transport-based counterfactuals [32, 96] | PN, PS, PNS [6, 80, 82, 103, 127]; response functions framework / canonical partitioning [6, 97, 112, 137, 141, 142, 143, 147]; causal marginal problem [46, 115]; deep twin networks [129] | Variance of the treatment effects [2]; CDF of the joint potential outcomes distribution [37]; conservative counterfactual effects [4] | **CSM (this paper)** |
| | SM | | ETT [119] ; path-specific effects [122, 146] ; deep generative models [31, 84, 134, 135, 140] ; semi-Markovian BGMs [93] | | Future work (out of the scope of this paper) | Future work (see discussion in Appendix E); ANMs with hidden confounding [69] |

*Legend:*
● M/SM: Markovian SCM (M), semi-Markovian SCM (SM)

**1. Symbolic identifiability and point identification with probabilistic expressions.** Complete and sound algorithms were proposed for symbolic (non-parametric) identifiability of the counterfactuals based on observational ($\mathcal{L}_1$) or interventional ($\mathcal{L}_2$) information, and causal diagram of a semi-Markovian SCM. For example, [121] adapted d-separation for the parallel-worlds networks [3] for simple counterfactual queries, and [27] extended it to nested counterfactuals with counterfactual unnesting theorem. These methods aim to provide a probabilistic expression, in the case the query is identifiable, which is then used in downstream point identification methods. Rather rare examples of identifiable queries include the treatment effect of the treated (ETT) [119] and path-specific effects [122, 146] for certain SCMs, e. g., when the treatment is binary. If the query is non-identifiable, like in the case with ECOU [ECOT] as in our paper, no guidance is provided on how to perform partial identification and methods can *not* be used.

**2(i). Point counterfactual identification with SCMs.** Another way to perform a point identification when the counterfactual query is symbolically non-identifiable is to restrict a functional class in SCMs. Examples of restricted Markovian SCMs include nonlinear additive noise models (ANMs) [66, 123], post nonlinear models (PNL) [148], location-scale noise models (LSNMs) [59], and bijective generation mechanisms (BGMs) [93, 94]. The latest, BGMs, also work in semi-Markovian settings with additional assumptions. Numerous deep generative models were also proposed for

point identification in both Markovian and semi-Markovian settings. Examples are normalizing flows [100]; diffusion models [21, 113]; variational inference [70, 100, 114, 134, 135]; adversarial learning [29, 84, 100, 117, 118, 140], and neural expectation-maximization [31]. As it was shown in [94], all those deep generative models above need to assume the BGM of the data to yield valid counterfactual inference.[4] The assumption of monotonicity of BGMs effectively sets the dimensionality of the latent noise to the same as the observed variables, which is rarely a realistic assumption. In our paper, we relax this assumption, and let the latent noise have arbitrary dimensionality.

**2(ii). Point counterfactual identification with non-SCMs-based approaches.** Alternative, non-SCM-based definitions of the counterfactuals exist, e. g., transport-based counterfactuals [32, 96], and conservative counterfactual effects [4]. Still, for the identifiability of ECOU [ECOT], they also rely on additional assumptions such as monotonicity.[5] These approaches were shown to coincide with standard SCM-based counterfactuals when the latent noise can be deterministically defined with observed data. Therefore, they coincide with BGMs.

**3(i). Discrete partial counterfactual identification.** Partial counterfactual identification was rigorously studied only for discrete SCMs or discrete outcomes. For example, bounds under no assumptions were derived for counterfactual probabilities [6, 80, 82, 103, 127]. More general counterfactual queries can also be tackled with response functions framework [6] and, more generally, canonical partitioning [112] in combination with deep neural networks [97, 137, 141, 142, 143, 147]. The same idea was used for causal marginal problem [46, 115] where the authors combined different experimental and observational data with overlapping observed variables. However, response functions framework and canonical partitioning can *not* be extended to the continuous setting in practice, as their computational complexity grows exponentially with respect to the cardinality of the observed variables. Hence, the aforementioned methods are *not* relevant baselines in our setting.

**3(ii). Continuous partial counterfactual identification of queries with informative bounds.** Certain counterfactual queries have ignorance intervals with informative bounds, which have a closed-form solution using Frechet-Hoeffding bounds for copulas. For example, informative bounds were derived, e. g., for the variance of the treatment effects [2], $\mathrm{Var}(Y_a - Y_{a'})$; for the CDF of the joint potential outcomes distribution [37], $\mathbb{P}(Y_a, Y_{a'})$, and other related queries [4]. Yet, these bounds cannot be applied to our setting, as our query of interest, namely, ECOU [ECOT], has *non-informative* bounds and, thus, requires a *sensitivity model*.

**3(iii). Continuous partial interventional identification.** The problem of partial interventional identification arises in semi-Markovian SCMs and usually aims at hidden confounding issues of treatment effect estimation [88]. For example, instrumental variable (IV) models always obtain informative bounds under the assumption of instrumental validity [48], so that partial identification is formulated as an optimization problem [55, 68, 99, 145]. In other cases, hidden confounding causes non-informative bounds and additional assumptions about its strength are needed, i. e., a *sensitivity model*. For example, the marginal sensitivity model (MSM) assumes the odds ratio between nominal and true propensity scores and derives informative bounds for average or conditional average treatment effects (ATE and CATE, respectively) [15, 35, 40, 41, 60, 61, 62, 89, 98, 126]. The outcome sensitivity models [15, 106] and confounding functions framework [17, 109] were also introduced for the sensitivity analysis of ATE and CATE. [49] develops a sensitivity model for ATE assuming a noise level of proxy variables. All the mentioned sensitivity models operate on conditional distributions and, therefore, do not extend to counterfactual partial identification, as the latter requires assumptions about the SCM. Some methods indeed restrict functional classes in SCMs to achieve informative bounds. For example, linear SCMs are assumed for partial identification of average treatment derivative [5]. By developing our CSM in our paper, we discuss what restrictions are required for SCMs to achieve informative bounds for partial counterfactual identification of continuous outcomes. Nevertheless, the methods for partial interventional identification are *not* aimed at counterfactual inference, and, thus, are *not* directly applicable in our setting.

---

[4]In semi-Markovian SCM additional assumptions are needed about the shared latent noise [31, 93].

[5]In the context of the non-SCM-based counterfactuals, the monotonicity assumption assumes a stochastic dominance of one potential outcome over the other [87, 103], i. e., $\mathbb{P}(Y_a \geq Y_{a'}) = 1$.

## A.2 Identifiability of latent variable models and disentanglement

The question of identifying latent noise variables was also studied from the perspective of nonlinear independent component analysis (ICA) [47, 57, 67]. Khemakhem et al. [67] showed that the joint distribution over observed and latent noise variables is in general unidentifiable. Although, nonlinear ICA was applied for interventional inference [133], these works did *not* consider SCMs or counterfactual queries.

# B Background materials

**Geometric measure theory.** Let $\delta(x) : \mathbb{R} \to \mathbb{R}$ be a Dirac delta function, defined as zero everywhere, except for $x = 0$, where it has a point mass of 1. Dirac delta function induces a Dirac delta measure, so that (with the slight abuse of notation)

$$\int_{\mathbb{R}} f(x)\,\delta(\mathrm{d}x) = \int_{\mathbb{R}} f(x)\,\delta(x)\,\mathrm{d}x = f(0), \tag{12}$$

where $f$ is a $C^0$ function with compact support. Dirac delta function satisfies the following important equality

$$\int_{\mathbb{R}} f(x)\,\delta(g(x))\,\mathrm{d}x = \sum_i \frac{f(x_i)}{|g'(x_i)|}, \tag{13}$$

where $f$ is a $C^0$ function with compact support, $g$ is a $C^1$ function, $g'(x_i) \neq 0$, and $x_i$ are roots of the equation $g(x) = 0$.

In addition, we define the $s$-dimensional Hausdorff measure $\mathcal{H}^s$, as in [95]. Let $E \subseteq \mathbb{R}^n$ be a $s$-dimensional smooth manifold ($s \leq n$) embedded into $\mathbb{R}^n$. Then, $E$ is a Borel subset in $\mathbb{R}^n$, is Hausdorff-measurable, and $s$-dimensional Hausdorff measure $\mathcal{H}^s(E)$ is the $s$-dimensional surface volume of $E$. For example, if $s = 1$, the Hausdorff measure coincides with a line integral, and, if $s = 2$, with surface integral:

$$s = 1: \quad E = \begin{pmatrix} x_1(t) \\ \vdots \\ x_n(t) \end{pmatrix} = \mathbf{x}(t) \quad \Rightarrow \quad \mathrm{d}\mathcal{H}^1(\mathbf{x}) = \left\| \frac{\mathrm{d}\mathbf{x}(t)}{\mathrm{d}t} \right\|_2 \mathrm{d}t, \tag{14}$$

$$s = 2: \quad E = \begin{pmatrix} x_1(s,t) \\ \vdots \\ x_n(s,t) \end{pmatrix} = \mathbf{x}(s,t) \quad \Rightarrow \quad \mathrm{d}\mathcal{H}^2(\mathbf{x}) = \left\| \frac{\mathrm{d}\mathbf{x}(s,t)}{\mathrm{d}s} \times \frac{\mathrm{d}\mathbf{x}(s,t)}{\mathrm{d}t} \right\|_2 \mathrm{d}s\,\mathrm{d}t, \tag{15}$$

where $\times$ is a vector product, $x_i(t)$ is a $t$-parametrization of the line, and $x_i(s,t)$ is a $(s,t)$-parametrization of the surface.

Also, $\mathcal{H}^n(E) = \mathrm{vol}^n(E)$ for Lebesgue measurable subsets $E \subseteq \mathbb{R}^n$, where $\mathrm{vol}^n$ is a standard $n$-dimensional volume (Lebesgue measure). In the special case of $s = 0$, Hausdorff measure is a counting measure: $\int_E f(x)\,\mathrm{d}\mathcal{H}^0(x) = \sum_{x \in E} f(x)$.

Importantly, the Hausdorff measure is related to the high-dimensional Dirac delta measure via the coarea formula [53] (Theorem 6.1.5), [95] (Theorem 3.3):

$$\int_{\mathbb{R}^n} f(\mathbf{x})\delta(g(\mathbf{x}))\,\mathrm{d}\mathbf{x} = \int_{E:\{g(\mathbf{x})=0\}} \frac{f(\mathbf{x})}{\|\nabla_\mathbf{x} g(\mathbf{x})\|_2}\,\mathrm{d}\mathcal{H}^{n-1}(\mathbf{x}), \tag{16}$$

where $f$ is $C^0$ function with compact support, and $g$ is a $C^1$ function with $\nabla_\mathbf{x} g(\mathbf{x}) > 0$ for $\mathbf{x} \in E$. Functions for which $\nabla_\mathbf{x} g(\mathbf{x}) > 0$ for $\mathbf{x} \in \mathbb{R}^d$ holds are called *regular*.

We define a *bundle of level sets* of the function $y = f(x)$ as a set of the level sets, indexed by $y$, i.e., $\{E(y); y \in \mathcal{Y} \in \mathbb{R}\}$, where $E(y) = \{y \in \mathbb{R} : f(x) = y\}$ and $\mathcal{Y} \subseteq \mathbb{R}$. A bundle of level sets of regular functions are closely studied in the *Morse theory*. In particular, we further rely on the *fundamental Morse theorem*, i.e., the level set bundles of a regular function are diffeomorphic to each other. This means that there exists a continuously differentiable bijection between every pair of the bundles.

**Differential geometry of manifolds.** In the following, we formally define the notion of the curvature for level sets. As the result of the *implicit function theorem*, level sets $E(y)$ of a regular function $f$ of class $C^1$ are Riemannian manifolds [78], namely smooth differentiable manifolds. Riemannian manifolds can be locally approximated via Euclidean spaces, i.e., they are equipped with a tangent space $T_\mathbf{x}(E(y))$ at a point $\mathbf{x}$, and a dot product defined on the tangent space. The tangent space, $T_\mathbf{x}(E(y))$, is orthogonal to the normal of the manifold, $\nabla_\mathbf{x} f(\mathbf{x})$.

For the level sets of regular functions of the class $C^2$, we can define the so-called *curvature* [43, 78]. Informally, curvature defines the extent a Riemannian manifold bends in different directions.

Convex regions of the manifold correspond to a negative curvature (in all directions), and concave regions to positive curvature, respectively. Saddle points have curvatures of different signs in different directions. Formally, the curvature is defined via the rate of change of the unit normal of the manifold, which is parameterized with the orthogonal basis $\{\tilde{x}_1, \ldots, \tilde{x}_{n-1}\}$ of the tangent space, $T_{\mathbf{x}}(E(y))$. This rate of change, namely a differential, forms a *shape operator* (or second fundamental form) on the tangent space, $T_{\mathbf{x}}(E(y))$. Then, eigenvalues of the shape operator are called *principal curvatures*, $\kappa_i(\mathbf{x})$ for $i \in \{1, \ldots, n-1\}$. Principal curvatures are a measure of the extrinsic curvature, i. e., the curvature of the manifold with respect to the embedding space, $\mathbb{R}^n$.

Principal curvatures for Riemannian manifolds, defined as the level sets of a regular $C^2$ function $f$, can be also expressed via the gradient and the Hessian of the following function [43]:

$$\kappa_i(\mathbf{x}) = -\frac{\operatorname{root}_i \left\{ \det \begin{pmatrix} \operatorname{Hess}_{\mathbf{x}} f(\mathbf{x}) - \lambda I & \nabla_{\mathbf{x}} f(\mathbf{x}) \\ (\nabla_{\mathbf{x}} f(\mathbf{x}))^T & 0 \end{pmatrix} = 0 \right\}}{\|\nabla_{\mathbf{x}} f(\mathbf{x})\|_2}, \quad i \in \{1, \ldots, n-1\}, \quad (17)$$

where $\operatorname{root}_i$ are roots of the equation with respect to $\lambda \in \mathbb{R}$. For $\mathbf{x} \in \mathbb{R}^2$, the level sets $E(y)$ are curves, and there is only one principal curvature

$$\kappa_1(\mathbf{x}) = -\frac{1}{2} \nabla_{\mathbf{x}} \left( \frac{\nabla_{\mathbf{x}} f(\mathbf{x})}{\|\nabla_{\mathbf{x}} f(\mathbf{x})\|_2} \right). \quad (18)$$

One of the important properties of the principal curvatures is that we can locally parameterize the manifold as the second-order hypersurface, i. e.,

$$y = \frac{1}{2}(\kappa_1 \tilde{x}_1^2 + \cdots + \kappa_{n-1} \tilde{x}_{n-1}^2) + O(\|\tilde{\mathbf{x}}\|_2^3), \quad (19)$$

where $\{\tilde{x}_1, \ldots, \tilde{x}_{n-1}\}$ is the orthogonal basis of the tangent space $T_{\mathbf{x}}(E(y))$.

## C Examples

**Example 1** (Counterfactual non-identifiability in Markovian-SCMs (continued))**.** *Here, we continue the example and provide the inference of observational (interventional) and counterfactual distributions for the SCMs $\mathcal{M}_1$ and $\mathcal{M}_2$.*

*Let us consider $\mathcal{M}_1$ first. It is easy to see that a pushforward of uniform distribution in a unit square with $f_Y(A, U_{Y^1}, U_{Y^2}) = A(U_{Y^1} - U_{Y^2} + 1) + (1 - A)(U_{Y^1} + U_{Y^2} - 1)$ induces triangular distributions. For example, for $f_Y(0, U_{Y^1}, U_{Y^2})$, a cumulative distribution function (CDF) of $\mathbb{P}^{\mathcal{M}_1}(Y \mid A = 0) = \mathbb{P}^{\mathcal{M}_1}(Y_{a=0})$ will have the form*

$$\mathbb{F}^{\mathcal{M}_1}(y \mid A = 0) = \mathbb{P}^{\mathcal{M}_1}(Y \leq y \mid A = 0) = \mathbb{P}(U_{Y^1} + U_{Y^2} - 1 \leq y) \tag{20}$$

$$= \begin{cases} 0, & (y \leq -1) \vee (y > 1), \\ \int_{\{u_{Y^1} + u_{Y^2} - 1 \leq y\}} \mathrm{d}u_{Y^1}\, \mathrm{d}u_{Y^2}, & \textit{otherwise} \end{cases} \tag{21}$$

$$= \begin{cases} 0, & (y \leq -1) \vee (y > 1), \\ \frac{(y+1)^2}{2}, & y \in (-1, 0], \\ 1 - \frac{(1-y)^2}{2}, & y \in (0, 1], \end{cases} \tag{22}$$

*which is the CDF of a triangular distribution. Analogously, a CDF of $\mathbb{P}^{\mathcal{M}_1}(Y \mid A = 1) = \mathbb{P}^{\mathcal{M}_1}(Y_{a=1})$ is*

$$\mathbb{F}^{\mathcal{M}_1}(y \mid A = 1) = \mathbb{P}^{\mathcal{M}_1}(Y \leq y \mid A = 1) = \mathbb{P}(U_{Y^1} - U_{Y^2} + 1 \leq y) \tag{23}$$

$$= \begin{cases} 0, & (y \leq 0) \vee (y > 2), \\ \int_{\{u_{Y^1} - u_{Y^2} + 1 \leq y\}} \mathrm{d}u_{Y^1}\, \mathrm{d}u_{Y^2}, & \textit{otherwise} \end{cases} \tag{24}$$

$$= \begin{cases} 0, & (y \leq 0) \vee (y > 2), \\ \frac{y^2}{2}, & y \in (0, 1], \\ 1 - \frac{(2-y)^2}{2}, & y \in (1, 2]. \end{cases} \tag{25}$$

*To infer the counterfactual outcome distribution of the untreated, $\mathbb{P}^{\mathcal{M}_1}(Y_{a=1} \mid A' = 0, Y' = 0)$, we make use of Lemma 2 and properties of the Dirac delta function (e. g., Eq. (13)). We yield*

$$\mathbb{P}^{\mathcal{M}_1}(Y_{a=1} = y \mid A' = 0, Y' = 0) = \int_{\{u_{Y^1} + u_{Y^2} - 1 = 0\}} \frac{\delta(u_{Y^1} - u_{Y^2} + 1 - y)}{\|\nabla_{u_Y}(u_{Y^1} + u_{Y^2} - 1)\|_2}\, \mathrm{d}\mathcal{H}^1(u_Y) \tag{26}$$

$$\overset{(*)}{=} \frac{1}{\sqrt{2}} \int_0^2 \sqrt{\left(\frac{1}{2}\right)^2 + \left(\frac{1}{2}\right)^2}\, \delta(t - y)\, \mathrm{d}t = \frac{1}{2} \int_0^2 \delta(t - y)\, \mathrm{d}t = \begin{cases} \frac{1}{2}, & y \in [0, 2], \\ 0, & \textit{otherwise}, \end{cases} \tag{27}$$

*where $(*)$ introduces a parametrization of the line $\{u_{Y^1} + u_{Y^2} - 1 = 0\}$ with $t$, namely*

$$\begin{pmatrix} u_{Y^1} \\ u_{Y^2} \end{pmatrix} = \begin{pmatrix} \frac{1}{2}t \\ 1 - \frac{1}{2}t \end{pmatrix}, \quad t \in [0, 2] \quad \Rightarrow \quad \mathrm{d}\mathcal{H}^1(u_Y) = \sqrt{\left(\frac{1}{2}\right)^2 + \left(\frac{1}{2}\right)^2}\, \mathrm{d}t. \tag{28}$$

*Therefore, the ECOU for $\mathcal{M}_1$ is*

$$Q_{0 \to 1}^{\mathcal{M}_1}(0) = \int_0^2 \frac{1}{2}\, y\, \mathrm{d}y = 1. \tag{29}$$

*Now, let us consider $\mathcal{M}_2$. Hence, $f_Y(0, U_{Y^1}, U_{Y^2})$ is the same for both $\mathcal{M}_1$ and $\mathcal{M}_2$, so are the observational (interventional) distributions, i. e., $\mathbb{P}^{\mathcal{M}_1}(Y \mid A = 0) = \mathbb{P}^{\mathcal{M}_2}(Y \mid A = 0)$. The same is true for $f_Y(1, U_{Y^1}, U_{Y^2})$ with $(0 \leq U_{Y^1} \leq 1) \wedge (U_{Y^1} \leq U_{Y^2} \leq 1)$ or, equivalently, $\mathbb{P}^{\mathcal{M}_1}(Y = y \mid A = 1) = \mathbb{P}^{\mathcal{M}_2}(Y = y \mid A = 1)$ with $y \in (0, 1]$.*

*Now, it is left to check whether $\mathbb{P}^{\mathcal{M}_2}(Y = y \mid A = 1)$ is the density of a triangular distribution for $y \in (1, 2]$. For that, we define level sets of the function $f_Y(1, U_{Y^1}, U_{Y^2})$ in the remaining part of the unit square in a specific way. Formally, they are a family of "bent" lines in the transformed two-dimensional space $\tilde{u}_{Y^1}, \tilde{u}_{Y^2}$ with*

$$\tilde{u}_{Y^2} = 8t^2\, |\tilde{u}_{Y^1}| + b(t), \quad t \in [0, 1]; \tilde{u}_{Y^1} \in [-1, 1]; \qquad \tilde{u}_{Y^2} \in [0, 1 - |\tilde{u}_{Y^1}|], \tag{30}$$

*where $b(t)$ is a bias, depending on $t$. The area under the line should change with quadratic speed, as it will further define the CDF of the induced observational distribution, $\mathbb{F}^{\mathcal{M}_2}(y \mid A = 1), y \in (1, 2]$, i. e.*

$$S(t) = 2t - t^2, \tag{31}$$

*so that $S(0) = 0$ and $S(1) = 1$. On the other hand, the area under the line from the family is*

$$S(t) = 2 \int_0^{\tilde{u}_*} (8t^2 \tilde{u}_{Y^1} + b(t)) \, \mathrm{d}\tilde{u}_{Y^1} + (1 - \tilde{u}_*)^2 = 8t^2 \tilde{u}_*^2 + 2\tilde{u}_* b(t) + (1 - \tilde{u}_*)^2, \tag{32}$$

*where $\tilde{u}_* = \frac{1 - b(t)}{8t^2 + 1}$. Therefore, we can find the dependence of $b(t)$ on $t$, so that the area $S(t)$ is preserved:*

$$2t - t^2 = 8t^2 \tilde{u}_*^2 + 2\tilde{u}_* b(t) + (1 - \tilde{u}_*)^2 \iff b(t) = 1 - \sqrt{(t - 1)^2 (8t^2 + 1)}. \tag{33}$$

*Let us reparametrize $t$ with $t = y - 1$, so that the line from a family with $t = 0$ corresponds to $y = 1$ and $t = 1$ to $y = 2$, respectively. We then yield*

$$\tilde{u}_{Y^2} = 2(y - 1) |\tilde{u}_{Y^1}| + 1 - \sqrt{(y - 2)^2 (8(y - 1)^2 + 1)}. \tag{34}$$

*In order to obtain the function $f_Y(1, U_{Y^1}, U_{Y^2})$ in the original coordinates $u_{Y^1}, u_{Y^2}$, we use a linear transformation $T$ given by*

$$T : \begin{pmatrix} \tilde{u}_{Y^1} \\ \tilde{u}_{Y^2} \end{pmatrix} \rightarrow \begin{pmatrix} -u_{Y^1} - u_{Y^2} + 1 \\ u_{Y^1} - u_{Y^2} \end{pmatrix}. \tag{35}$$

*Hence, the family of "bent" lines can be represented as the following implicit equation:*

$$F(y, u_{Y^1}, u_{Y^2}) = u_{Y^1} - u_{Y^2} - 2(y - 1) |-u_{Y^1} - u_{Y^2} + 1| - 1 + \sqrt{(y - 2)^2 (8(y - 1)^2 + 1)} = 0. \tag{36}$$

*Importantly, the determinant of the Jacobian of the transformation $T$ is equal to 2; therefore, the area under the last line $S(1)$ shrinks from 1 (in space $\tilde{u}_{Y^1}, \tilde{u}_{Y^2}$) to 0.5 (in space $u_{Y^1}, u_{Y^2}$). Thus, we can also easily verify with Eq. (31) that the CDF of the induced observational distribution coincides with the CDF of the triangular distribution for $y \in (1, 2]$:*

$$\mathbb{F}^{\mathcal{M}_2}(y \mid A = 1) = \frac{1}{2} + \mathbb{P}^{\mathcal{M}_2}(1 \leq Y \leq y \mid A = 1) = \frac{1}{2} + \frac{1}{2} S(y - 1) = 1 - \frac{(2 - y)^2}{2}. \tag{37}$$

*To infer the counterfactual outcome distribution of the untreated, i.e., $\mathbb{P}^{\mathcal{M}_2}(Y_{a=1} \mid A' = 0, Y' = 0)$, we again use the Lemma 2 and properties of Dirac delta function (e. g., Eq. (13)). We yield*

$$\mathbb{P}^{\mathcal{M}_2}(Y_{a=1} = y \mid A' = 0, Y' = 0) = \frac{1}{2}, \quad y \in (0, 1], \tag{38}$$

$$\mathbb{P}^{\mathcal{M}_2}(Y_{a=1} = y \mid A' = 0, Y' = 0) = \int_{\{u_{Y^1} + u_{Y^2} - 1 = 0\}} \frac{\delta(F(y, u_{Y^1}, u_{Y^2}))}{\|\nabla_{u_Y}(u_{Y^1} + u_{Y^2} - 1)\|_2} \, \mathrm{d}\mathcal{H}^1(u_Y) \tag{39}$$

$$\stackrel{(*)}{=} \frac{1}{\sqrt{2}} \int_0^1 \sqrt{\left(\frac{1}{2}\right)^2 + \left(\frac{1}{2}\right)^2} \delta\left(t - 1 + \sqrt{(y - 2)^2 (8(y - 1)^2 + 1)}\right) \mathrm{d}t \tag{40}$$

$$= \frac{1}{2} \left| -\left(\sqrt{(y - 2)^2 (8(y - 1)^2 + 1)}\right)' \right| = \frac{(5 - 4y)^2 (2 - y)}{2\sqrt{(y - 2)^2 (8(y - 1)^2 + 1)}}, \quad y \in (1, 2], \tag{41}$$

*where $(*)$ introduces a parametrization of the line $\{u_{Y^1} + u_{Y^2} - 1 = 0\}$ with $t$, namely*

$$\begin{pmatrix} u_{Y^1} \\ u_{Y^2} \end{pmatrix} = \begin{pmatrix} \frac{1}{2} t + \frac{1}{2} \\ \frac{1}{2} - \frac{1}{2} t \end{pmatrix}, \quad t \in [0, 1] \quad \Rightarrow \quad \mathrm{d}\mathcal{H}^1(u_Y) = \sqrt{\left(\frac{1}{2}\right)^2 + \left(\frac{1}{2}\right)^2} \, \mathrm{d}t. \tag{42}$$

*Finally, the ECOU for $\mathcal{M}_2$ can be calculated numerically via*

$$Q_{0 \to 1}^{\mathcal{M}_2}(0) = \int_0^1 \frac{1}{2} y \, \mathrm{d}y + \int_1^2 \frac{(5 - 4y)^2 (2 - y)}{2\sqrt{(y - 2)^2 (8(y - 1)^2 + 1)}} \, \mathrm{d}y \approx \frac{1}{4} + 0.864 \approx 1.114. \tag{43}$$

**Example 2** (Box-Müller transformation). *Here, we demonstrate the application of the Lemma 1 to infer the standard normal distribution with the Box-Müller transformation. The Box-Müller transformation is a well-established approach to sample from the standard normal distribution, which omits the usage of the inverse CDF of the normal distribution. Formally, the Box-Müller transformation is described by an SCM $\mathcal{M}_{\mathrm{bm}}$ of class $\mathfrak{B}(C^1, 2)^6$ with the following function for $Y$:*

$$\mathcal{M}_{\mathrm{bm}} : f_Y(A, U_{Y^1}, U_{Y^2}) = f_Y(U_{Y^1}, U_{Y^2}) = \sqrt{-2\log(U_{Y^1})}\,\cos(\pi U_{Y^2}). \tag{44}$$

*We will now use the Lemma 1 to verify that $\mathbb{P}^{\mathcal{M}_{\mathrm{bm}}}(Y \mid a) = N(0, 1)$. We yield*

$$\mathbb{P}^{\mathcal{M}_{\mathrm{bm}}}(Y = y \mid a) = \int_{\{\sqrt{-2\log(u_{Y^1})}\,\cos(\pi u_{Y^2})=y\}} \frac{1}{\left\|\nabla_{u_Y}\left(\sqrt{-2\log(u_{Y^1})}\,\cos(\pi u_{Y^2})\right)\right\|_2} \mathrm{d}\mathcal{H}^1(u_Y)$$
$$\tag{45}$$

$$= \int_{\{\sqrt{-2\log(u_{Y^1})}\,\cos(\pi u_{Y^2})=y\}} \frac{1}{\sqrt{-\frac{\cos^2(\pi u_{Y^2})}{2\,u_{Y^1}^2\,\log(u_{Y^1})} - 2\,\pi^2\log(u_{Y^1})\sin^2(\pi u_{Y^2})}} \mathrm{d}\mathcal{H}^1(u_Y) \tag{46}$$

$$\overset{(*)}{=} \int_0^{1/2} \frac{\sqrt{1 + \left(-\frac{\pi y^2 \sin(\pi t)}{\cos^3(\pi t)}\exp\left(-\frac{y^2}{2\cos^2(\pi t)}\right)\right)^2}}{\sqrt{\frac{\cos^4(\pi t)}{y^2}\exp\left(\frac{y^2}{\cos^2(\pi t)}\right) + \frac{\pi^2\sin^2(\pi t)y^2}{\cos^2(\pi t)}}} \mathrm{d}t = \int_0^{1/2} \frac{|y|\exp\left(-\frac{y^2}{2\cos^2(\pi t)}\right)}{\cos^2(\pi t)} \mathrm{d}t \tag{47}$$

$$= \frac{1}{\sqrt{2\pi}}\exp\left(-\frac{y^2}{2}\right) = N(y; 0, 1), \tag{48}$$

*where $(*)$ introduces a parametrization of the level set $\{\sqrt{-2\log(u_{Y^1})}\,\cos(\pi u_{Y^2}) = y\}, y > 0$ with $t$, namely*

$$\begin{pmatrix} u_{Y^1} \\ u_{Y^2} \end{pmatrix} = \begin{pmatrix} \exp\left(-\frac{y^2}{2\cos^2(\pi t)}\right) \\ t \end{pmatrix}, \quad t \in [0, \tfrac{1}{2}) \tag{49}$$

$$\Rightarrow \quad \mathrm{d}\mathcal{H}^1(u_Y) = \sqrt{\left(-\frac{\pi y^2\sin(\pi t)}{\cos^3(\pi t)}\exp\left(-\frac{y^2}{2\cos^2(\pi t)}\right)\right)^2 + 1}\,\mathrm{d}t. \tag{50}$$

*This parametrization is also valid for $y < 0$ and $t \in (\frac{1}{2}, 1]$. Due to the symmetry of the function $f_Y$, we only consider one of the cases (see Fig. 8 (left) with the level sets).*

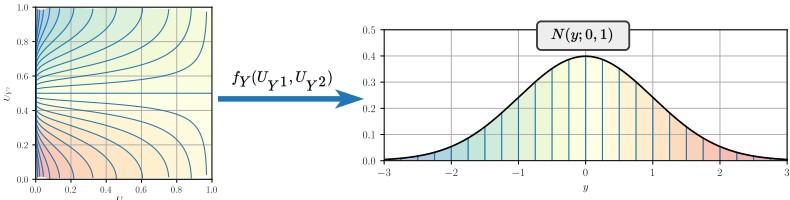

Figure 8: Box-Müller transformation as an example of the SCM of class $\mathfrak{B}(C^1, 2)$ (see Example 2). Here, $N(y; 0, 1)$ is the density of the standard normal distribution.

---

[6]Formally, $f_Y \in C^1$ only for $u_Y \in (0, 1)^2$.

**Example 3** (Connected components of the factual level sets). *Here, we construct the function with the level sets consisting of multiple connected components. For that, we extend Example 2 with the Box-Müller transformation. We define a so-called oscillating Box-Müller transformation*

$$f_Y(U_{Y^1}, U_{Y^2}) = \sqrt{-2\log(U_{Y^1})}\cos(2^{-\lceil\log_2(U_{Y^2})\rceil}\pi U_{Y^2}). \tag{51}$$

*We plot the level set for $y = -0.5$ in Fig. 9, namely $E(y) = \{u_Y \in [0,1]^2 : f_Y(u_{Y^1}, u_{Y^2}) = y\}$. Here, we see that the level sets consist of an infinite number of connected components with an infinite total length.*

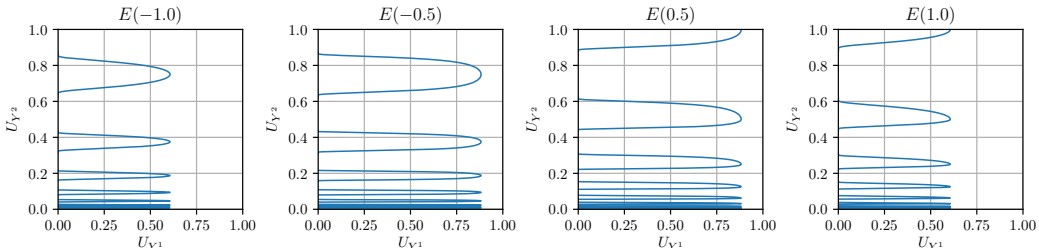

Figure 9: Level sets with multiple connected components $E(y) = \{u_Y \in [0,1]^2 : f_Y(u_{Y^1}, u_{Y^2}) = y\}$ for oscillating Box-Müller transformation (see Example 3).

**Example 4** (Curvature of level sets). *Let us consider three SCMs of class $\mathfrak{B}(C^2, 2)$, which satisfy the Assumption $\kappa$ with $\kappa = 50$. Then, the curvature of the level sets is properly defined for the function $f_Y$ (see Eq. (11)). Fig. 10 provides a heatmap with the absolute curvature, $|\kappa_1(u_Y)|$, for the counterfactual level sets, $\{E(y, a) : y \in \mathcal{Y}_a\}$. Here, all three SCMs are instances of our APID.*

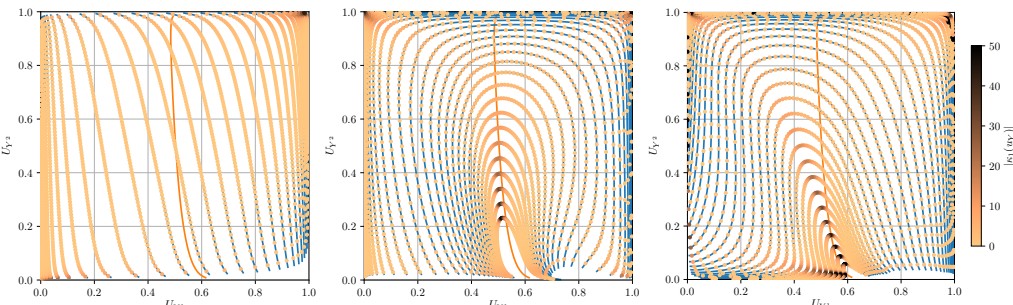

Figure 10: Curvature of the counterfactual level sets $\{E(y, a) : y \in \mathcal{Y}_a\}$ in blue for three SCMs, satisfying the Assumption $\kappa$. Factual level set for the quantile of the distribution, $E(\mathbb{F}_{a'}^{-1}(0.5), a')$ is given for reference in orange. As can be seen, "bent" level sets have larger curvature (right) than "straight" (left). Here, $\kappa = \max_{y \in \mathcal{Y}_a, u_Y \in E(a,y)} |\kappa_1(u_Y)|$ amounts to 50.

**Example 5** ($\mathcal{M}_{\text{tri}}$). *There exist BGMs in class $\mathfrak{B}(C^1, 1)$, for which the curvature of level sets are not properly defined at certain points. For example, an SCM $\mathcal{M}_{\text{tri}}$, which induces triangular distributions, like in Example 1. For this example, $\mathcal{M}_{\text{tri}}$ will have the following function for $Y$:*

$$\mathcal{M}_{\text{tri}} : f_Y(A, U_Y) = \begin{cases} A\sqrt{2U_Y} + (A-1)(\sqrt{2U_Y} - 1), & \text{if } U_Y \in [0.0, 0.5], \\ A(2 - \sqrt{2(U_Y+1)}) + (A-1)(1 - \sqrt{2(U_Y+1)}), & \text{if } U_Y \in (0.5, 1.0]. \end{cases} \tag{52}$$

**Example 6** ($\mathcal{M}_{\text{flat}}$). *Here, we provide examples of SCMs in the class $\mathfrak{B}(C^\infty, d)$, for which the bundles of the level sets coincide for both treatments, i.e., $\{E(y, a) : y \in \mathcal{Y}_a\} = \{E(y, a') : y \in \mathcal{Y}_{a'}\}$, and all the level sets are flat hypersurfaces ($\kappa = 0$). For example, SCMs with the following functions for $Y$:*

$$\mathcal{M}_{\text{flat}} : f_Y(A, U_Y) = g(A, w_1 U_{Y^1} + \cdots + w_d U_{Y^d}), \tag{53}$$

*where $g(a, \cdot)$ is an invertible function in class $C^\infty$ and $w_1, \ldots, w_d$ are coefficients from the linear combination. After a reparametrization, it is always possible to find an equivalent BGM with the function:*

$$g(A, w_1 U_{Y^1} + \cdots + w_d U_{Y^d}) = g(A, \tilde{U}_Y) = g(A, \mathbb{F}_{\tilde{U}_Y}^{-1}(U_{Y^1})), \tag{54}$$

where $\mathbb{F}_{\tilde{U}_Y}^{-1}(\cdot)$ is an inverse CDF of $\tilde{U}_Y = w_1 U_{Y^1} + \cdots + w_d U_{Y^d}$. *The smoothness of the inverse CDF then defines the smoothness of the BGM.*

**Example 7** ($\mathcal{M}_{\mathrm{curv}}$). *This example is the generalization of the Example 6. Here, we also use the same bundles of the level sets for both treatments, but allow for the curvature of $\kappa > 0$:*

$$\mathcal{M}_{\mathrm{curv}} : f_Y(A, U_Y) = g(A, h(U_Y)), \tag{55}$$

*where $g(a, \cdot)$ is an invertible function in class $C^\infty$, and $h(\cdot) : [0, 1]^d \to \mathbb{R}$ is a function in class $C^\infty$ with bounded by $\kappa$ curvature of the level sets. In this case, we also can reparametrize the latent space with a non-linear transformation and find an equivalent BGM:*

$$g(A, h(U_Y)) = g(A, \tilde{U}_Y) = g(A, \mathbb{F}_{\tilde{U}_Y}^{-1}(U_{Y^1})), \tag{56}$$

*where $\mathbb{F}_{\tilde{U}_Y}^{-1}(\cdot)$ is an inverse CDF of $\tilde{U}_Y = h(U_Y)$. The smoothness of the inverse CDF then analogously defines the smoothness of the BGM.*

**Example 8** ($\mathcal{M}_{\mathrm{perp}}$). *We can construct an SCM of class $\mathfrak{B}(C^\infty, 2)$, so that the level sets are all straight lines ($\kappa = 0$) and perpendicular to each other for different $a \in \{0, 1\}$, e. g.,*

$$\mathcal{M}_{\mathrm{perp}} : f_Y(A, U_{Y^1}, U_{Y^2}) = A\, g_1(U_{Y^1}) + (A - 1)\, g_2(U_{Y^2}), \tag{57}$$

*where $g_1(\cdot), g_2(\cdot)$ are invertible functions in class $C^\infty$. In this case, the ECOU for $y' \in \mathcal{Y}_{a'}$ always evaluates to the conditional expectation*

$$Q_{0 \to 1}^{\mathcal{M}_{\mathrm{perp}}}(y') = \frac{1}{\mathbb{P}^{\mathcal{M}_{\mathrm{perp}}}(Y = y' \mid A = 0)} \int_{\{u_{Y^2} = g_2^{-1}(y')\}} \frac{g_1(u_{Y^1})}{\left|\nabla_{u_{Y^2}} g_2(u_{Y^2})\right|} \, \mathrm{d}\mathcal{H}^1(u_Y) \tag{58}$$

$$= \frac{\mathbb{P}^{\mathcal{M}_{\mathrm{perp}}}(Y = g_2(g_2^{-1}(y')) \mid A = 0)}{\mathbb{P}^{\mathcal{M}_{\mathrm{perp}}}(Y = y' \mid A = 0)} \int_0^1 g_1(u_{Y^1}) \, \mathrm{d}u_{Y^1} = \mathbb{E}^{\mathcal{M}_{\mathrm{perp}}}(Y \mid A = 1). \tag{59}$$

*Similarly, the ECOT is $\mathbb{E}^{\mathcal{M}_{\mathrm{perp}}}(Y \mid A = 0)$. This result is, in general, different from the result of BGMs, which, e. g., would yield $Q_{a' \to a}^{\mathcal{M}}(y') = \mathbb{F}_a(0.5)$ for $y' = \mathbb{F}_{a'}^{-1}(0.5)$. Thus, point identification is not guaranteed with $\kappa = 0$.*

## D  Proofs

**Uniformity of the latent noise.** Classes of SCMs $\mathfrak{B}(C^k, d)$ assume independent uniform latent noise variables, i.e., $U_Y \sim \text{Unif}(0,1)^d$. This assumption does not restrict the distribution of the latent noise. Namely, for a bivariate SCM $\mathcal{M}_{\text{non-unif}}$ with $d$-dimensional non-uniform continuous latent noise, it is always possible to find an equivalent[7] SCM $\mathcal{M}_{\text{unif}}$ of class $\mathfrak{B}(C^k, d)$. This follows from the change of variables formula and the inverse probability transformation [25, 116]:

$$\mathcal{M}_{\text{non-unif}} : Y = \tilde{f}_Y(A, \tilde{U}_Y), \qquad\qquad \tilde{U}_Y \sim \mathbb{P}(\tilde{U}_Y); \tilde{f}_Y(a, \cdot) \in C^k, \qquad (60)$$

$$\Longleftrightarrow \quad \mathcal{M}_{\text{unif}} : Y = \tilde{f}_Y(A, T(U_Y)) = f_Y(A, U_Y), \quad U_Y \sim \text{Unif}(0,1)^d; f_Y(a, \cdot) \in C^{\min(1,k)}, \qquad (61)$$

where $\tilde{U}_Y = T(U_Y)$, and $T(\cdot) : (0,1)^d \to \mathbb{R}^d$ is a diffeomorphism (bijective $C^1$ transformation) and a solution to the following equation:

$$\underbrace{\mathbb{P}(\tilde{U}_Y = \tilde{u}_Y)}_{\in C^0} = \left| \det \left( \frac{\mathrm{d} \overbrace{T^{-1}(\tilde{u}_Y)}^{\in C^1}}{\mathrm{d}\tilde{u}_Y} \right) \right|. \qquad (62)$$

For further details on the existence and explicit construction of $T(\cdot)$, see [25]. Thus, $f_Y(a, \cdot)$ is of class $C^{\min(1,k)}$, as the composition of $T(\cdot)$ and $\tilde{f}_Y(a, \cdot)$.

In our CSM we require the functions of the SCMs to be of class $C^2$. In this case, $\mathfrak{B}(C^2, d)$ also includes all the SCMs with non-uniform continuous latent noise with densities of class $C^1$. This also follows from the change of variables theorem, see Eq. (62).

More general classes of functions, e.g., all measurable functions, or more general classes of distributions, e.g., distributions with atoms, fall out of the scope of this paper, even though we can find bijective (but non-continuous) transformations between probability spaces of the same cardinality (see the isomorphism of Polish spaces, Theorem 9.2.2 in [12]).

**Lemma 1** (Observational distribution as a pushforward with $f_Y$). *Let $\mathcal{M} \in \mathfrak{B}(C^1, d)$. Then, the density of the observational distribution, induced by $\mathcal{M}$, is*

$$\mathbb{P}^{\mathcal{M}}(Y = y \mid a) = \int_{E(y,a)} \frac{1}{\|\nabla_{u_Y} f_Y(a, u_Y)\|_2} \, \mathrm{d}\mathcal{H}^{d-1}(u_Y), \qquad (1)$$

*where $E(y,a)$ is a level set (preimage) of $y$, i.e., $E(y,a) = \{u_Y \in [0,1]^d : f_Y(a, u_Y) = y\}$, and $\mathcal{H}^{d-1}(u_Y)$ is the Hausdorff measure (see Appendix B for the definition).*

*Proof.* Lemma is a result of the coarea formula (see Eq. (16)):

$$\mathbb{P}^{\mathcal{M}}(Y = y \mid a) = \int_{[0,1]^d} \mathbb{P}^{\mathcal{M}}(Y = y, U_Y = u_Y \mid a) \, \mathrm{d}u_Y \qquad (63)$$

$$\overset{(*)}{=} \int_{[0,1]^d} \mathbb{P}^{\mathcal{M}}(Y = y \mid u_Y, a) \, \mathbb{P}^{\mathcal{M}}(U_Y = u_Y) \, \mathrm{d}u_Y \qquad (64)$$

$$= \int_{[0,1]^d} \delta(f_Y(a, u_Y) - y) \, 1 \, \mathrm{d}u_Y = \int_{E(y,a)} \frac{1}{\|\nabla_{u_Y} f_Y(a, u_Y)\|_2} \, \mathrm{d}\mathcal{H}^{d-1}(u_Y), \qquad (65)$$

where $(*)$ holds, as $U_Y$ is independent of $A$. In a special case, when we set $d = 1$, we obtain a change of variables formula

$$\mathbb{P}^{\mathcal{M}}(Y = y \mid a) = \int_{E(y,a)} \frac{1}{|\nabla_{u_Y} f_Y(a, u_Y)|} \, \mathrm{d}\mathcal{H}^0(u_Y) = \sum_{u_Y \in E(y,a)} |\nabla_{u_Y} f_Y(a, u_Y)|^{-1}. \qquad (66)$$

$\square$

---

[7]Equivalence of the SCMs is defined as the almost surely equality of all the functions in two SCMs [13].

**Corollary 1** (Identifiable functions (BGMs)[93]). *The function $f_Y(a, u_Y)$ can be identified (up to a sign) given the observational distribution $\mathbb{P}(Y \mid a)$, induced by some SCM $\mathcal{M}$ of class $\mathfrak{B}(C^1, 1)$ if it is strictly monotonous in $u_Y$. In this case, the function is*

$$f_Y(a, u_Y) = \mathbb{F}_a^{-1}(\pm u_Y \mp 0.5 + 0.5), \tag{67}$$

*where $\mathbb{F}_a^{-1}$ is an inverse CDF of the observational distribution, and the sign switch corresponds to the strictly monotonically increasing and decreasing functions.*

*Proof.* As $d = 1$, we can apply the change of variables formula from Eq. (66):

$$\mathbb{P}(Y = y \mid a) = \sum_{u_Y \in E(y,a)} |\nabla_{u_Y} f_Y(a, u_Y)|^{-1} = |\nabla_{u_Y} f_Y(a, u_Y)|^{-1}, \quad u_Y = f_Y^{-1}(a, y), \tag{68}$$

where $f_Y^{-1}(a, y)$ is an inverse function with respect to $u_Y$ (it is properly defined, as the function is strictly monotonous). Thus, to find $f_Y$, we have to solve the following differential equation [116]:

$$\mathbb{P}(Y = f_Y(a, u_Y) \mid a) = |\nabla_{u_Y} f_Y(a, u_Y)|^{-1}. \tag{69}$$

By the properties of derivatives, this equation is equivalent to

$$\mathbb{P}(Y = y \mid a) = \left| \nabla_y f_Y^{-1}(a, y) \right|. \tag{70}$$

Since the latter holds for every $y$, we can integrate it from $-\infty$ to $f_Y(a, u_Y)$:

$$\int_{-\infty}^{f_Y(a,u_Y)} \mathbb{P}(Y = y \mid a) \, \mathrm{d}y = \int_{-\infty}^{f_Y(a,u_Y)} \left| \nabla_y f_Y^{-1}(a, y) \right| \mathrm{d}y \tag{71}$$

$$= \begin{cases} \int_{f_Y^{-1}(a,-\infty)}^{u_Y} \mathrm{d}t = \int_0^{u_Y} \mathrm{d}t = u_Y, & f_Y \text{ is strictly increasing,} \\ -\int_{f_Y^{-1}(a,-\infty)}^{u_Y} \mathrm{d}t = \int_{u_Y}^1 \mathrm{d}t = 1 - u_Y, & f_Y \text{ is strictly decreasing.} \end{cases} \tag{72}$$

Therefore, $\mathbb{F}_a(f_Y(a, u_Y)) = \pm u_Y \mp 0.5 + 0.5$. $\qquad\square$

**Corollary 2** (Critical points of $f_Y(a, \cdot)$). *Lemma 1 also implies that critical points of $f_Y(a, u_Y)$, i.e., $\{u_Y \in [0, 1]^d : \|\nabla_{u_Y} f_Y(a, u_Y)\|_2 = 0\}$, are mapped onto points $y$ with infinite density. Therefore, the assumption, that the function is regular, namely $\nabla_{u_Y} f_Y(a, u_Y) > 0$, is equivalent to the assumption of the continuous observational density with the finite density.*

**Lemma 2.** *Let $\mathcal{M} \in \mathfrak{B}(C^1, d)$. Then, the density of the counterfactual outcome distribution of the [un]treated is*

$$\mathbb{P}^{\mathcal{M}}(Y_a = y \mid a', y') = \frac{1}{\mathbb{P}^{\mathcal{M}}(Y = y' \mid a')} \int_{E(y',a')} \frac{\delta(f_Y(a, u_Y) - y)}{\|\nabla_{u_Y} f_Y(a', u_Y)\|_2} \, \mathrm{d}\mathcal{H}^{d-1}(u_Y), \tag{2}$$

*where $\delta(\cdot)$ is a Dirac delta function, and the expected counterfactual outcome of the [un]treated, i.e., ECOU [ECOT], is*

$$Q_{a' \to a}^{\mathcal{M}}(y') = \mathbb{E}^{\mathcal{M}}(Y_a \mid a', y') = \frac{1}{\mathbb{P}^{\mathcal{M}}(Y = y' \mid a')} \int_{E(y',a')} \frac{f_Y(a, u_Y)}{\|\nabla_{u_Y} f_Y(a', u_Y)\|_2} \, \mathrm{d}\mathcal{H}^{d-1}(u_Y), \tag{3}$$

*where $E(y', a')$ is a (factual) level set of $y'$, i. e., $E(y', a') = \{u_Y \in [0, 1]^d : f_Y(a', u_Y) = y'\}$ and $a' \neq a$.*

*Proof.* Both counterfactual queries can be inferred with the abduction-action-prediction procedure.

(1) Abduction infers the posterior distribution of the latent noise variables, conditioned on the evidence:

$$\mathbb{P}^{\mathcal{M}}(U_Y = u_Y \mid a', y') = \frac{\mathbb{P}^{\mathcal{M}}(U_Y = u_Y, A = a', Y = y')}{\mathbb{P}^{\mathcal{M}}(A = a', Y = y')} \overset{(*)}{=} \frac{\mathbb{P}^{\mathcal{M}}(Y = y' \mid a', u_Y) \, \mathbb{P}^{\mathcal{M}}(U_Y = u_Y)}{\mathbb{P}^{\mathcal{M}}(Y = y' \mid a')} \tag{73}$$

$$= \frac{\delta(f_Y(a', u_Y) - y')}{\mathbb{P}^{\mathcal{M}}(Y = y' \mid a')}, \tag{74}$$

where $(*)$ holds, as $U_Y$ is independent of $A$.

(2)-(3) Action and prediction are then pushing forward the posterior distribution with the counterfactual function. For example, the density of the counterfactual outcome distribution of the [un]treated is

$$\mathbb{P}^{\mathcal{M}}(Y_a = y \mid a', y') = \int_{[0,1]^d} \mathbb{P}^{\mathcal{M}}(Y_a = y, U_Y = u_Y \mid a', y') \, du_Y \tag{75}$$

$$= \int_{[0,1]^d} \mathbb{P}^{\mathcal{M}}(Y_a = y \mid a', y', u_Y) \, \mathbb{P}^{\mathcal{M}}(U_Y = u_Y \mid a', y') \, du_Y \tag{76}$$

$$\overset{(*)}{=} \frac{1}{\mathbb{P}^{\mathcal{M}}(Y = y' \mid a')} \int_{[0,1]^d} \delta(f_Y(a, u_Y) - y)\, \delta(f_Y(a', u_Y) - y')\, du_Y \tag{77}$$

$$= \frac{1}{\mathbb{P}^{\mathcal{M}}(Y = y' \mid a')} \int_{E(y', a')} \frac{\delta(f_Y(a, u_Y) - y)}{\|\nabla_{u_Y} f_Y(a', u_Y)\|_2} \, d\mathcal{H}^{d-1}(u_Y), \tag{78}$$

where $(*)$ holds due to the independence of $Y$ from $Y'$ and $A'$, conditional on $U_Y$ (see the parallel worlds network in Fig. 1). The inference of the ECOU [ECOT] is analogous

$$Q^{\mathcal{M}}_{a' \to a}(y') = \mathbb{E}^{\mathcal{M}}(Y_a \mid a', y') = \int_{\mathcal{Y}_a} \int_{[0,1]^d} \mathbb{P}^{\mathcal{M}}(Y_a = y, U_Y = u_Y \mid a', y') \, du_Y \, dy \tag{79}$$

$$= \int_{[0,1]^d} \mathbb{E}^{\mathcal{M}}(Y_a \mid a', y', u_Y) \, \mathbb{P}^{\mathcal{M}}(U_Y = u_Y \mid a', y') \, du_Y \tag{80}$$

$$\overset{(*)}{=} \frac{1}{\mathbb{P}^{\mathcal{M}}(Y = y' \mid a')} \int_{[0,1]^d} f_Y(a, u_Y)\, \delta(f_Y(a', u_Y) - y')\, du_Y \tag{81}$$

$$= \frac{1}{\mathbb{P}^{\mathcal{M}}(Y = y' \mid a')} \int_{E(y', a')} \frac{f_Y(a, u_Y)}{\|\nabla_{u_Y} f_Y(a', u_Y)\|_2} \, d\mathcal{H}^{d-1}(u_Y), \tag{82}$$

where $(*)$ holds due to the independence of $Y$ from $Y'$ and $A'$, conditional on $U_Y$ (see the parallel worlds network in Fig. 1), and due to the law of the unconscious statistician. $\qquad\square$

**Corollary 3** (Identifiable counterfactuals (BGMs bounds)[93])**.** *The ECOU [ECOT] can be identified (up to a sign) given the observational distribution $\mathbb{P}(Y \mid a)$, induced by some SCM $\mathcal{M}$ of class $\mathfrak{B}(C^1, 1)$ if it is strictly monotonous in $u_Y$. In this case, the ECOU [ECOT] is*

$$Q^{\mathcal{M}}_{a' \to a}(y') = \mathbb{F}_a^{-1}(\pm \mathbb{F}_{a'}(y') \mp 0.5 + 0.5). \tag{83}$$

*Proof.* The corollary is a result of the the Lemma 2 and the Corollary 1:

$$Q^{\mathcal{M}}_{a' \to a}(y') = \frac{1}{\mathbb{P}^{\mathcal{M}}(Y = y' \mid a')} \int_{E(y', a')} \frac{f_Y(a, u_Y)}{|\nabla_{u_Y} f_Y(a', u_Y)|} \, d\mathcal{H}^0(u_Y) \tag{84}$$

$$= |\nabla_{u_Y} f_Y(a', u_Y)| \, \frac{f_Y(a, u_Y)}{|\nabla_{u_Y} f_Y(a', u_Y)|} = f_Y(a, u_Y), \quad u_Y = f_Y^{-1}(a', y'), \tag{85}$$

Therefore, the ECOU [ECOT] is

$$Q^{\mathcal{M}}_{a' \to a}(y') = \mathbb{F}_a^{-1}(\pm u_Y \mp 0.5 + 0.5) = \mathbb{F}_a^{-1}(\pm \mathbb{F}_{a'}(y') \mp 0.5 + 0.5). \tag{86}$$

$\qquad\square$

**Theorem 1** (Non-informative bounds of ECOU (ECOT))**.** *Let the continuous observational distribution $\mathbb{P}(Y \mid a)$ be induced by some SCM of class $\mathfrak{B}(C^\infty, d)$. Let $\mathbb{P}(Y \mid a)$ have a compact support $\mathcal{Y}_a = [l_a, u_a]$ and be of finite density $\mathbb{P}(Y = y \mid a) < +\infty$. Then, the ignorance interval for the partial identification of the ECOU [ECOT] of class $\mathfrak{B}(C^\infty, d)$, $d \geq 2$, has non-informative bounds: $\underline{Q_{a' \to a}(y')} = l_a$ and $\overline{Q_{a' \to a}(y')} = u_a$.*

*Proof.* Without the loss of generality, let us consider the lower bound of the ECOU [ECOT], namely,

$$\underline{Q_{a' \to a}(y')} = \inf_{\mathcal{M} \in \mathfrak{B}(C^k, d)} Q^{\mathcal{M}}_{a' \to a}(y') \quad \text{s.t. } \forall a \in \{0, 1\} : \mathbb{P}(Y \mid a) = \mathbb{P}^{\mathcal{M}}(Y \mid a). \tag{87}$$

The proof then proceeds in two steps. First, we prove the statement of the theorem for $d = 2$, i. e., when latent noise is two-dimensional. Then, we extend it to arbitrary dimensionality.

*Step 1 ($d = 2$).* Lemma 2 suggests that to minimize the ECOU [ECOT], we have to either increase the proportion of the length (one-dimensional volume) of the factual level set, which intersects the bundle of counterfactual level sets, or change the counterfactual functions, by "bending" the bundle of counterfactual level sets around a factual level set. Here, we focus on the second case, which is sufficient to construct an SCM with non-informative bounds.

Formally, we can construct a sequence of SCMs $\{\mathcal{M}_{\text{non-inf}}^{\varepsilon(y)} : y \in \mathcal{Y}_a, 0 < \varepsilon(y) < 1\}$ of class $\mathfrak{B}(C^\infty, 2)$, for which $Q_{a' \to a}^{\mathcal{M}_{\text{non-inf}}^{\varepsilon(y)}}(y')$ gets arbitrarily close to $l_a$ as $y \to l_a$ and $\varepsilon(y) \to 0$.

For all $\{\mathcal{M}_{\text{non-inf}}^{\varepsilon(y)} : y \in \mathcal{Y}_a, 0 < \varepsilon(y) < 1\}$, we choose the same factual function

$$f_Y(a', U_{Y^1}, U_{Y^2}) = \mathbb{F}_{a'}^{-1}(U_{Y^2}), \tag{88}$$

where $\mathbb{F}_{a'}^{-1}(\cdot)$ is the inverse CDF of the observational distribution $\mathbb{P}(Y \mid a')$. This is always possible as a result of the Corollaries 1 and 2. Hence, all the level sets of the factual function are horizontal straight lines of length 1 (see Fig. 4), due to $E(y', a') = \{\mathbb{F}_{a'}^{-1}(u_{Y^2}) = y'\}$.

Now, we construct the counterfactual functions in the following way. For a fixed $\varepsilon(y)$, we choose the level sets of the function $f_Y(a, \cdot)$, $\{E^{\varepsilon(y)}(y, a), y \in \mathcal{Y}_a\}$, so that they satisfies the following properties. (1) Each $E^{\varepsilon(y)}(y, a)$ intersects the factual level set $E(y', a')$ at a certain point once and only once and ends at the boundary of the boundaries of the unit square. (2) Each $E^{\varepsilon(y)}(y, a)$ splits the unit square on two parts, an interior with the area $\mathbb{F}_a(y)$ and an exterior, with the area $1 - \mathbb{F}_a(y)$, respectively (where $\mathbb{F}_a(\cdot)$ is the CDF of the counterfactual distribution). This property ensures that the induced CDF coincides with the observational CDF at every point, namely, $\forall y \in \mathcal{Y}_a : \mathbb{F}_a^{\mathcal{M}_{\text{non-inf}}^{\varepsilon(y)}}(y) = \mathbb{F}_a(y)$. (3) All the level sets have the nested structure, namely, all the points of the interior of $E^{\varepsilon(y)}(y, a)$ are mapped to $\underline{y} < y$, and points of the exterior, to $\overline{y} > y$. Additionally, for some fixed $y \in \mathcal{Y}_a$, we set $\varepsilon = \varepsilon(y)$ as the proportion of the factual level set, $E(y', a')$, fully contained in the exterior of the counterfactual level set, $E(y, a)$. Thus, the interior of the counterfactual level set covers $1 - \varepsilon(y)$ of the factual level set.

Therefore, the ECOU [ECOT] for $\mathcal{M}_{\text{non-inf}}^{\varepsilon(y)}$ for some fixed $y \in \mathcal{Y}_a$ is

$$Q_{a' \to a}^{\mathcal{M}_{\text{non-inf}}^{\varepsilon(y)}}(y') = (1 - \varepsilon(y))\underline{y} + \varepsilon(y)\overline{y}, \quad \underline{y} \in [l_a, y], \overline{y} \in [y, u_a], \tag{89}$$

which follows from the mean value theorem for integrals, since, for every $y \in \mathcal{Y}_a$, we can choose $\varepsilon(y)$ arbitrarily close to zero, and, thus

$$\inf_{y \in \mathcal{Y}_a, 0 < \varepsilon(y) < 1} Q_{a' \to a}^{\mathcal{M}_{\text{non-inf}}^{\varepsilon(y)}}(y') = \inf_{y \in \mathcal{Y}_a, 0 < \varepsilon(y) < 1} (1 - \varepsilon(y))\underline{y} + \varepsilon(y)\overline{y} = l_a. \tag{90}$$

*Step 2 ($d > 2$).* The construction of the factual and counterfactual level sets extends straightforwardly to the higher dimensional case. In this case, the factual level sets are straight hyperplanes and the counterfactual level sets are hypercylinders, "bent" around the factual hyperplanes. $\qquad\square$

**Theorem 2** (Informative bounds with our CSM). *Let the continuous observational distribution $\mathbb{P}(Y \mid a)$ be induced by some SCM of class $\mathfrak{B}(C^2, d), d \geq 2$, which satisfies Assumption $\kappa$. Let $\mathbb{P}(Y \mid a)$ have a compact support $\mathcal{Y}_a = [l_a, u_a]$. Then, the ignorance interval for the partial identification of ECOU [ECOT] of class $\mathfrak{B}(C^2, d)$ has informative bounds, dependent on $\kappa$ and $d$, which are given by $\underline{Q_{a' \to a}(y')} = l(\kappa, d) > l_a$ and $\overline{Q_{a' \to a}(y')} = u(\kappa, d) < u_a$.*

*Proof.* We will now show that ECOU [ECOT] always have informative bounds for every possible SCM satisfying the assumptions of the theorem. Without the loss of generality, let us consider the lower bound, i. e.,

$$\underline{Q_{a' \to a}(y')} = \inf_{\mathcal{M} \in \mathfrak{B}(C^k, d)} Q_{a' \to a}^{\mathcal{M}}(y') \quad \text{s.t. } \forall a \in \{0, 1\} : \mathbb{P}(Y \mid a) = \mathbb{P}^{\mathcal{M}}(Y \mid a). \tag{91}$$

The proof contains three steps. First, we show that the level sets of $C^2$ functions with compact support consist of the countable number of connected components, each with the finite Hausdorff

measure. Each connected component of almost every level set is thus a $d-1$-dimensional Riemannian manifold. Second, we demonstrate that, under Assumption $\kappa$, almost all the level set bundles have a nested structure, are diffeomorphic to each other, have boundaries, and their boundaries always coincide with the boundary of the unit hypercube $[0,1]^d$. Third, we arrive at a contradiction in that, to obtain non-informative bounds, we have to fit a $d$-ball of arbitrarily small radius to the interior of the counterfactual level set (which is not possible with bounded absolute principal curvature). Therefore, CSM with Assumption $\kappa$ has informative bounds.

*Step 1.* The structure of the level sets of $C^2$ functions with the compact support (or more generally, Lipschitz functions) were studied in [1]. We employ Theorem 2.5 from [1], which is a result of Sard's theorem. Specifically, the level sets of $C^2$ functions with the compact support consist of the countable number of connected components, each with the finite Hausdorff measure (surface volume), namely, $\mathcal{H}^{d-1}(E(y,a)) < \infty$ for $a \in \{0,1\}$. In Example 3, we provide an SCM with the level sets with indefinitely many connected components. Furthermore, connected components of almost every level set are $(d-1)$-dimensional Riemannian manifolds of class $C^2$. Some points $y$ have $(d-2)$- and lower dimensional manifolds, but their probability measure is zero (e. g., the level sets of the bounds of the support, $E(l_a, a)$ and $E(u_a, a)$).

*Step 2.* The Assumption $\kappa$ assumes the existence of the principal curvatures at every point of the space (except for the boundaries of the unit hypercube $[0,1]^d$), for both the factual and the counterfactual function. Thus, the functions of the SCMs are regular, as otherwise the principal curvatures would not be defined at the critical points (see Eq. (17)). As a result of the fundamental Morse theorem (see Appendix B), almost all the level set bundles are nested and diffeomorphic to each other. Specifically, the level set bundles $\{E(y,a) : y \in (l_a, u_a)\}$ for $a \in \{0,1\}$. The latter exclude the level sets of the bounds of the $\mathcal{Y}_a$, i. e., $E(l_a, a)$ and $E(u_a, a)$ for $a \in \{0,1\}$, which in turn are laying completely in the boundary of the unit hypercube $[0,1]^d$. Another important property is that all the level sets have a boundary (otherwise, due to the diffeomorphism, the critical point would exist in their interior) and this boundary lies at the boundary of the unit hypercube $[0,1]^d$.

*Step 3.* Let us fix the factual level set, $E(y', a')$. By the observational distribution constraint, we know that the interior occupies $\mathbb{F}_{a'}(y')$ of the total volume of the unit hypercube $[0,1]^d$, where $\mathbb{F}_{a'}(\cdot)$ is the CDF of the observational distribution. Also, its absolute principal curvatures are bounded by $\kappa$.

Then, let us assume there exists a counterfactual function, which obtains non-informative bounds of ECOU [ECOT]. For this, a counterfactual level set for some $y$ close to $l_a$ has to contain exactly $F_a(y)$ volume in the interior. At the same time, this level set has to contain as much of the surface volume of the factual level set, $\mathcal{H}^{d-1}(E(y', a'))$, as possible in its interior. This is only possible by "bending" the counterfactual level set along one of the directions, so that the boundaries of the counterfactual level set lay at the boundaries of the unit hypercube, $[0,1]^d$. Due to the bound on the maximal absolute principal curvature, we have to be able to fully contain a $d$-ball with radius $\frac{1}{\kappa}$ inside the interior of the counterfactual level set. This $d$-ball occupies the volume

$$\text{vol}^d(d\text{-ball}) = \frac{\pi^{n/2}}{\Gamma(\frac{d}{2}+1)} \frac{1}{\kappa^d}, \tag{92}$$

where $\Gamma(\cdot)$ is a Gamma function. Therefore, the volume of the interior of the counterfactual level set has to be at the same time arbitrarily close to zero, as $F_a(y) \to 0$ as $y \to l_a$, and at least of the volume of the $d$-ball with radius $\frac{1}{\kappa}$. Thus, we arrived at the contradiction, which proves the theorem.

$\square$

**Corollary 4** (Monotonicity wrt. to $\kappa$). *By construction, it is guaranteed that, with decreasing $\kappa$, the bounds for ECOU [ECOT] could only shrink, i. e., for $\kappa_1 > \kappa_2 : l(\kappa_1, d) \leq l(\kappa_2, d)$ and $u(\kappa_1, d) \geq u(\kappa_2, d)$. This happens, as we decrease the class of functions in the constraints of the optimization problem from Definition 2.*

**Corollary 5** (Sufficiency of $d = 2$). *In general, upper and lower bounds under assumed CSM ($\kappa$) are different for different $d_1, d_2 \geq 2$, e. g., $l(\kappa, d_1) \leq l(\kappa, d_2)$ and $u(\kappa, d_1) \geq u(\kappa, d_2)$ for $d_1 > d_2$. This follows from the properties of the high-dimensional spaces, e. g., we one can fit more $d$-balls of the fixed radius to the hypercube $[0,1]^d$ with the increase of $d$. Nevertheless, without the loss of generality and in the absence of the information on the dimensionality, we can set $d = 2$, as we still cover the entire identifiability spectrum by varying $\kappa$.*

**Lemma 3** (BGMs-EQTDs identification gap of CSM($\kappa = 0$)). *Let the assumptions of Theorem 2 hold and let $\kappa = 0$. Then the ignorance intervals for the partial identification of ECOU [ECOT] of class $\mathfrak{B}(C^2, d)$ are defined by min/max between BGMs bounds and expectations of quantile-truncated distributions (EQTDs):*

$$\underline{Q_{a' \to a}(y')}/\overline{Q_{a' \to a}(y')} = \min / \max \{BGM_+(y'), BGM_-(y'), EQTD_l(y'), EQTD_u(y')\} \quad (7)$$

$$BGM_+(y') = \mathbb{F}_a^{-1}(\mathbb{F}_{a'}(y')) \qquad BGM_-(y') = \mathbb{F}_a^{-1}(1 - \mathbb{F}_{a'}(y')), \quad (8)$$

$$EQTD_l(y') = \mathbb{E}\left(Y \mid a, Y < \mathbb{F}_a^{-1}(1 - 2\left|0.5 - \mathbb{F}_{a'}(y')\right|)\right), \quad (9)$$

$$EQTD_u(y') = \mathbb{E}\left(Y \mid a, Y > \mathbb{F}_a^{-1}(2\left|\mathbb{F}_{a'}(y') - 0.5\right|)\right), \quad (10)$$

*where $\mathbb{E}(Y \mid Y < \cdot), \mathbb{E}(Y \mid Y > \cdot)$ are expectations of truncated distributions.*

*Proof.* We provide an informal proof for $d = 2$ based on geometric intuition. When $\kappa = 0$, all the level sets are straight lines (hyperplanes for $d > 2$). It is possible to characterize all the possible relative arrangements of the factual level set and the bundle of counterfactual level sets, and the corresponding values of ECOU [ECOT]. For a particular arrangement, displayed in Fig. 11, we see that the area of the interior of the factual level set is $\text{vol}^2 = \mathbb{F}_{a'}(y')$. At the same time, the factual level set crosses the counterfactual bundle corresponding to the interval $[l_a, \mathbb{F}_a^{-1}(2\mathbb{F}_{a'}(y'))]$, thus the counterfactual distribution of [un]treated is simply a $2\mathbb{F}_{a'}(y')$-quantile-truncated distribution, i.e., $\mathbb{P}(Y \mid a, Y < \mathbb{F}_a^{-1}(2\mathbb{F}_{a'}(y')))$. Therefore, the bounds of the ignorance interval of ECOU [ECOT] are defined by the min/max of both BGMs and expectations of quantile-truncated distributions (EQTDs). □

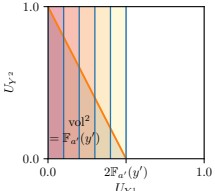

Figure 11: Factual level set (orange), counterfactual level sets (blue).

# E   Discussion of extensions and limitations

**Sharp bounds.** To obtain the sharp bounds under CSM ($\kappa$), one has to exactly solve the constrained variational problem task formulated in the Definition 2 in the space of the SCMs, which satisfy the Assumption $\kappa$. This is a very non-trivial task, the distributional constraint and the constraint of level sets of bounded curvature both include the partial differential equation with the Hausdorff integrals. For this task, in general, an explicit solution does not exist in a closed form (unlike, e. g., the solution of the marginal sensitivity model [126]).

**Extension of CSM to discrete treatments and continuous covariates.** Our CSM ($\kappa$) naturally scales to bivariate SCMs with categorical treatments, $A \in \{0, \ldots, K\}$; and multivariate Markovian SCMs with additional covariates, $X \in \mathbb{R}^m$, which are all predecessors of $Y$. These extensions, nevertheless, bring additional challenges from estimating $\mathbb{P}(Y \mid A, X)$ from the observational (interventional) data. Hence, additional smoothness assumptions are required during modeling.

**Extension of CSM to semi-Markovian SCMs.** CSM ($\kappa$) is limited to the Markovian SCMs. For semi-Markovian SCMs, e. g., a setting of the potential outcomes framework [111], additional assumptions are needed. Specifically, in semi-Markovian SCMs, the latent noise variables could be shared for $X$ and $Y$, and this complicates counterfactual inference. Future work may consider incorporating other sensitivity models to our CSM ($\kappa$), e. g., marginal sensitivity model, MSM ($\Gamma$) [126], as described in Fig. 12.

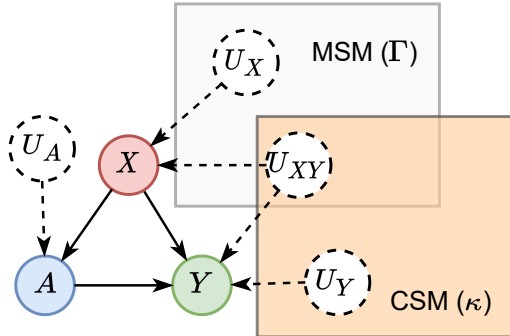

Figure 12: Possible combination of our CSM ($\kappa$) with the marginal sensitivity model, MSM ($\Gamma$) [126], for the potential outcome framework [111].

# F  Details on architecture and inference of *Augmented Pseudo-Invertible Decoder*

Our APID provides an implementation of CSM ($\kappa$) for SCMs of class $\mathfrak{B}(C^2, 2)$. In the following, we list several important requirements, that the underlying probabilistic model has to satisfy. Then, we explain how deep normalizing flows [107, 125] meet these requirements.

1. A probabilistic model has to explicitly model an arbitrary function $\hat{f}_Y(a, u_Y), u_Y \in [0, 1]^2$, of class $C^2$. Normalizing flows with *free-form Jacobians* can implement arbitrary invertible $C^2$ transformations from $\hat{F}_a : \mathbb{R}^2 \to \mathbb{R}^2$. Then, by omitting one of the outputs, we can model functions $f_Y(a, \cdot) : [0, 1]^2 \to \mathcal{Y}_a \subset \mathbb{R}$. We discuss normalizing flows with free-form Jacobians in Sec. F.1.

2. A model has to fit the observational (interventional) distribution, given a sample from it, $\mathcal{D} = \{A_i, Y_i\}_{i=1}^n \sim \mathbb{P}(A, Y)$. This is possible for the normalizing flows, as they are maximizing the *log-likelihood* of the data, $\hat{\mathbb{P}}(Y = Y_i \mid A = A_i)$, directly with the gradient-based methods. Importantly, we also employ variational augmentations [22] to evaluate the log-likelihood. We discuss this in detail in Sec. F.2.

3. A probabilistic model should be able to perform the estimation of ECOU [ECOT], through abduction-action-prediction steps. In normalizing flows, this can be achieved with the help of the *variational augmentations* [22]. Specifically, for evidence, $y'$ and $a'$, our proposed APID can infer arbitrarily many points from the estimated factual level set $\hat{E}(y', a')$, which follow the estimated posterior distribution of the latent noise, i.e., $\hat{\mathbb{P}}(U_Y \mid y', a')$. We discuss this in detail in Sec. F.2.

4. All the level sets of the modeled function need to have a bounded curvature. This is possible for normalizing flows as a deep learning model. Namely, the curvature $\kappa_1(u_Y)$ can be evaluated via *automatic differentiation* tools and added the loss of the model. We discuss this in detail in Sec. F.3.

## F.1  Choice of a normalizing flow with free-form Jacobians

Normalizing flows (NFs) [107, 125] differ in how flexible the modeled transformations in high dimensions are. Some models, like planar NF [107] or Sylvester NF [128], only allow for transformations with low-rank Jacobians. Other models employ masked auto-regressive networks [72] or coupling blocks [33, 34] to construct transformations with lower-triangular (structured) Jacobians.

Recently, several models were proposed for modeling free-form Jacobian transformations, e.g., (i) continuous NFs [23, 45] or (ii) residual NFs [9, 24]. However, this flexibility comes at a computational cost. For (i) continuous NFs, we have to solve a system of ordinary differential equations (ODEs) for every forward and reverse transformation. As noted by [45], numerical ODE solvers only work well for the non-stiff differential equations (=for non-steep transformations). For (ii) residual NFs, the computational complexity stems from the evaluation of the determinant of the Jacobian, which is required for the log-likelihood, and, from the reverse transformation, where we have to employ fixed point iterations.

We experimented with both continuous NFs and residual NFs but we found the latter to be more stable. The drawbacks of residual NFs can be partially fixed under our CSM ($\kappa$). First, the determinant of the Jacobian can be evaluated exactly, as we set $d = 2$. Second, the numerical complexity of the reverse transformation[8] can be lowered by adding more residual layers, so that each layer models less steep transformation and the total Lipschitz constant is larger for the whole transformation. Hence, in our work, we resorted to residual normalizing flows [9, 24].

## F.2  Variational augmentations and pseudo-invertibility

Variational augmentations were proposed for increasing the expressiveness of normalizing flows [22]. They augment the input to a higher dimension and then employ the invertible transformation of

---

[8]Note that, in our APID, forward transformation corresponds to the inverse function, i.e., $\hat{f}_Y(a, \cdot)$.

the flow. Hence, normalizing flows with variational augmentations can be seen as pseudo-invertible probabilistic models [10, 54].

We use variational augmentations to model the inverse function, $\hat{f}_Y(a, \grave{)}$, i.e., sample points from the level sets. For this, we augment the (estimated) outcome $Y \in \mathcal{Y}$ with $Y_{\text{aug}} \sim N(g^a(Y), \varepsilon^2) \in \mathbb{R}$, where $g^a(\cdot)$ is a fully-connected neural network with one hidden layer and parameters $\theta_a$, and $\varepsilon^2$ is a hyperparameter. As such, our proposed APID models $\hat{f}_Y(a, \cdot)$ through a two-dimensional transformation $\hat{F}_a = (\hat{f}_{Y_{\text{aug}}}(a, \cdot), \hat{f}_Y(a, \cdot)) : [0, 1]^2 \to \mathbb{R} \times \mathcal{Y}_a$. Variational augmentations facilitate our task in two ways.

(i) They allow evaluating the log-likelihood of the data via

$$\log \hat{\mathbb{P}}_{\beta_a, \theta_a}(Y = Y_i \mid a) = \mathbb{E}_{Y_{\text{aug}, i}} \left[ \log \hat{\mathbb{P}}_{\beta_a}(Y_{\text{aug}} = Y_{\text{aug}, i}, Y = Y_i) - \log N(Y_{\text{aug}, i}; g^a(Y_i), \varepsilon^2) \right],$$
(93)

where $\beta_a$ are the parameters of the residual normalizing flow, and $N(\cdot; g^a(Y_i), \varepsilon^2)$ is the density of the normal distribution. Here, we see that, by increasing the $\varepsilon^2$, the variance of the sample estimate of the log-likelihood of the data increases. On the other hand, for $\varepsilon^2 \to 0$, the transformations of the residual flow become steeper, as we have to transport the point mass to the unit square, $[0, 1]^2$.

(ii) Variational augmentations enable the abduction-action-prediction to estimate ECOU [ECOT] in a differential fashion. At the *abduction* step, we infer the sample from the posterior distribution of the latent noise, defined at the estimated factual level set $\hat{E}(y', a')$. For that, we variationally augment the evidence $y'$ with $b$ samples from $(y', \{Y'_{\text{aug}, j}\}_{j=1}^b \sim N(g^{a'}(y'), \varepsilon^2))$, and, then, transform them to the latent noise space with the factual flow

$$\{(\tilde{u}_{Y^1, j}, \tilde{u}_{Y^2, j})\}_{j=1}^b = \hat{F}_{a'}^{-1} \left( \{(y', Y'_{\text{aug}, j})\}_{j=1}^b \right).$$
(94)

Then, the *action* step selects the counterfactual flow, $\hat{F}_a(\cdot)$, and the *prediction* step transform the abducted latent noise with it:

$$\{(\hat{Y}_{\text{aug}, j}, \hat{Y}_j)\}_{j=1}^b = \hat{F}_a \left( \{(\tilde{u}_{Y^1, j}, \tilde{u}_{Y^2, j})\}_{j=1}^b \right).$$
(95)

In the end, ECOU [ECOT] is estimated by averaging $\hat{Y}_j$:

$$\hat{Q}_{a' \to a}(y') = \frac{1}{b} \sum_{j=1}^b \hat{Y}_j.$$
(96)

In our APID, the parameters of the residual normalizing flow, $\beta_a$, and the parameters of the variational augmentations, $\theta_a$, are always optimized jointly. We use the reparametrization trick, to back-propagate through sampling of the augmentations.

### F.3 Penalizing curvatures of the level sets

Although there exist deep learning models which explicitly bound the curvature of the modeled function (e.g., [124]), none of the works (to the best of our knowledge) did enforce the curvature of *the level sets* of the modeled function.

In our APID, the curvature $\kappa_1(u_Y)$ is evaluated using automatic differentiation exactly via Eq. (11) and then incorporated into the loss. As the calculation of the second derivatives is a costly operation, we heuristically evaluate the curvature only at the points of the counterfactual level set, which corresponds to the estimated ECOU [ECOT], namely $(u_{Y^1}, u_{Y^2}) \in \hat{E}(\hat{Q}_{a' \to a}(y'), a)$. For this, we again use the variationally augmented sample of size $b$ for $\hat{y} = \hat{Q}_{a' \to a}(y')$: $(\hat{y}, \{\hat{Y}_{\text{aug}, j}\}_{j=1}^b \sim N(g^a(\hat{y}), \varepsilon^2))$. Then, we use the counterfactual flow (such as in Eq. (94))

$$\{(\hat{u}_{Y^1, j}, \hat{u}_{Y^2, j})\}_{j=1}^b = \hat{F}_a^{-1} \left( \{(\hat{y}, \hat{Y}_{\text{aug}, j})\}_{j=1}^b \right),$$
(97)

while the curvature has to be only evaluated at $b$ points given by

$$\{\kappa_1(\hat{u}_{Y^1, j}, \hat{u}_{Y^2, j})\}_{j=1}^b.$$
(98)

# G  Details on training of *Augmented Pseudo-Invertible Decoder*

## G.1  Training objective

Our APID satisfies all the requirements set by CSM ($\kappa$), by combining several losses with different coefficients (see the overview in Fig. 13). Given observational data $\mathcal{D} = \{A_i, Y_i\}_{i=1}^n$ drawn i.i.d. and a counterfactual query, $Q_{a' \to a}(y')$, for the partial identification, APID minimizes the following losses:

**(1) Negative log-likelihood loss** aims to fit the observational data distribution by minimizing the negative log-likelihood of the data, $\mathcal{D}_a = \{A_i = a, Y_i\}$, modeled by APID. To prevent the overfitting, we use noise regularization [110], which adds a normally distributed noise: $\tilde{Y}_i = Y_i + \xi_i; \xi_i \sim N(0, \sigma^2)$, where $\sigma^2 > 0$ is a hyperparameter. Then, the negative log-likelihood loss for $a \in \{0, 1\}$ is

$$\mathcal{L}_{\text{NLL}}(\beta_a, \theta_a) = -\frac{1}{n} \sum_{i=1}^n \log \hat{\mathbb{P}}_{\beta_a, \theta_a}(Y = \tilde{Y}_i \mid a), \tag{99}$$

where the log-likelihood is evaluated according to the Eq. (93).

**(2) Wasserstein loss** prevents the posterior collapse [26, 132] of the APID, as the sample $\{Y_i\}$ is not guaranteed to cover the full latent noise space when mapped with the estimated inverse function. We thus sample $b$ points from the latent noise space, $\{(U_{Y^1, j}, U_{Y^2, j})\}_{j=1}^b \sim \text{Unif}(0, 1)^2$ and map with the forward transformation $F_a(\cdot)$ for $a \in \{0, 1\}$ via

$$\{(\hat{Y}_{\text{aug}, a, j}, \hat{Y}_{a, j})\}_{j=1}^b = \hat{F}_a \left(\{(U_{Y^1, j}, U_{Y^2, j})\}_{j=1}^b\right). \tag{100}$$

Then, we evaluate the empirical Wasserstein distance

$$\mathcal{L}_{\mathbb{W}}(\beta_a) = \int_0^1 |\hat{\mathbb{F}}_{\hat{Y}_a}^{-1}(q) - \hat{\mathbb{F}}_{Y_a}^{-1}(q)| \, dq, \tag{101}$$

where $\hat{\mathbb{F}}_{\hat{Y}}^{-1}(\cdot)$ and $\hat{\mathbb{F}}_Y^{-1}(\cdot)$ are empirical quantile functions, based on samples $\{A_i = a, Y_i\}$ and $\{\hat{Y}_{a, j}\}$, respectively.

**(3) Counterfactual query loss** aims to maximize/minimize ECOU [ECOT]:

$$\mathcal{L}_Q(\beta_{a'}, \theta_{a'}, \beta_a) = \text{Softplus}(\mp \hat{Q}_{a' \to a}(y')), \tag{102}$$

where $\mp$ changes for maximization/minimization correspondingly, $\hat{Q}_{a' \to a}(y')$ is evaluated as described in the Eq. (96), and $\text{Softplus}(x) = \log(1 + \exp(x))$. We use the softplus transformation to scale the counterfactual query logarithmically so that it matches the scale of the negative log-likelihood. We also block the gradients and fit only the counterfactual flow, when maximizing/minimizing ECOU [ECOT], to speed up the training.

**(4) Curvature loss** penalizes the curvature of the level sets of the modeled counterfactual function $\hat{f}_Y(a, \cdot)$:

$$\mathcal{L}_\kappa(\beta_a, \theta_a) = \frac{1}{b} \sum_{j=1}^b \kappa_1(\hat{u}_{Y^1, j}, \hat{u}_{Y^2, j}), \tag{103}$$

where $\{\kappa_1(\hat{u}_{Y^1, j}, \hat{u}_{Y^2, j})\}_{j=1}^b$ are defined in Sec. F.3.

All the losses are summed up to a single training objective

$$\mathcal{L}(\beta_{a'}, \theta_{a'}, \beta_a, \theta_a) = \sum_{a \in \{0, 1\}} [\mathcal{L}_{\text{NLL}}(\beta_a, \theta_a) + \mathcal{L}_{\mathbb{W}}(\beta_a)] + \lambda_Q \, \mathcal{L}_Q(\beta_{a'}, \theta_{a'}, \beta_a) + \lambda_\kappa \, \mathcal{L}_\kappa(\beta_a, \theta_a). \tag{104}$$

## G.2  Training algorithm and hyperparameters

**Training algorithm.** The training of APID proceeds in three stages; see the pseudocode in Algorithm 1. At a *burn-in stage* ($n_B = 500$ training iterations), we only fit two residual normalizing flows (one for each treatment) with $\mathcal{L}_{\text{NLL}}$ and $\mathcal{L}_{\mathbb{W}}$. Then, we copy the counterfactual flow and variational

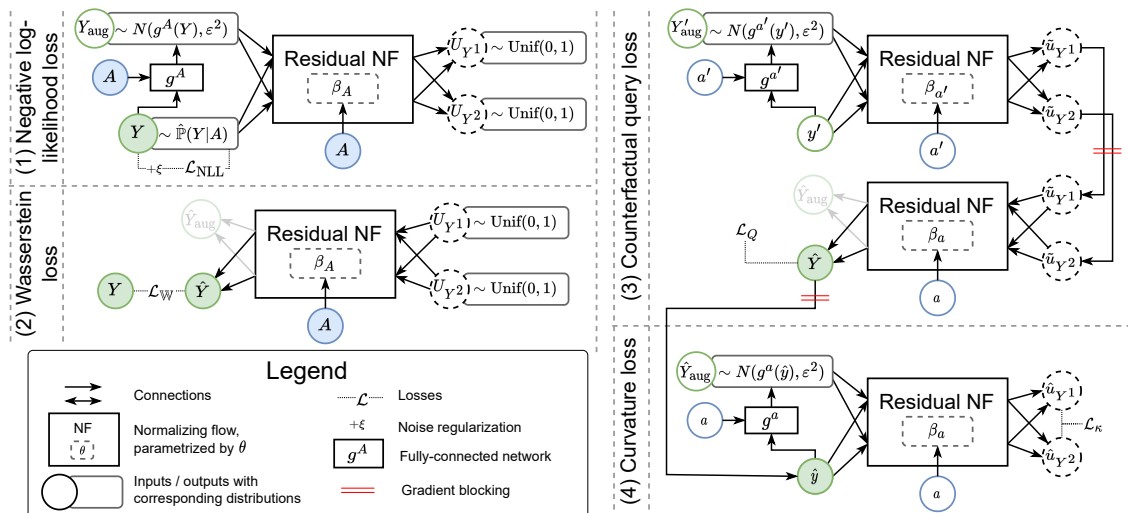

Figure 13: Overview of the training and inference of our *Augmented Pseudo-Invertible Decoder*.

augmentation parameters twice, for the task of maximization/minimization, respectively. Also, we freeze the factual flow parameters. At a *query stage* ($n_Q = 100$ training iterations), we, additionally to previous losses, enable the counterfactual query loss, $\lambda_Q \mathcal{L}_Q$. During this stage, two counterfactual flows are able to realign and start "bending" their level sets around the frozen factual level set. Ultimately, at the last *curvature-query stage* ($n_{CQ} = 500$ training iterations), all the parts of the loss in Eq. (104) are enabled. After training is over, we report the upper and lower bounds for the counterfactual query as an output of the APID with exponential moving average (EMA) of the model parameters [105]. EMA of the model parameters allows us to reduce the variance during training and is controlled via the smoothness hyperparameter $\gamma = 0.99$.

**Hyperparameters.** We use the Adam optimizer [71] with a learning rate $\eta = 0.01$ and a minibatch size of $b = 32$ to fit our APID. This $b = 32$ is also used for all the sampling routines at other losses. For the residual normalizing flows, we use $t = 15$ residual transformations, each with $h_t = 5$ units of the hidden layers. We set the relative and absolute tolerance of the fixed point iterations to $0.0001$ and a maximal number of iterations to $200$. For variational augmentations, we set the number of units in the hidden layer to $h_g = 5$, and the variance of the augmentation to $\varepsilon^2 = 0.5^2$. For the noise regularization, we chose $\sigma^2 = 0.001^2$. This configuration resulted in a good fit (with respect to the test log-likelihood) for all the experiments, and, thus, we did not tune hyperparameters. We varied the coefficients for the losses, $\lambda_Q$ and $\lambda_\kappa$, and report the results in Sec. 6 and Appendix H.

**Algorithm 1** Training algorithm of APID

---

1: **Input.** Counterfactual query $Q_{a' \to a}(y')$; training dataset $\mathcal{D} = \mathcal{D}_0 \cup \mathcal{D}_1$; number of training iterations $n_B, n_Q, n_{CQ}$; minibatch size $b$; learning rates $\eta$; intensity of the noise regularization $\sigma^2$; the variance of the augmentation $\varepsilon^2$; EMA smoothing $\gamma$; loss coefficients $\lambda_Q$ and $\lambda_\kappa$.

2:

3: **Init.** Flows parameters: residual NFs $(\beta_0, \beta_1)$ and variational augmentations $(\theta_0, \theta_1)$.

4: **for** $k = 0$ to $n_B$ **do**                                                ▷ **Burn-in stage**

5:     **for** $a \in \{0, 1\}$ **do**

6:         $\{A_i = a, Y_i\}_{i=1}^b \leftarrow$ minibatch of size $b$ from $\mathcal{D}_a$

7:         $\xi_i \sim N(0, \sigma^2); \tilde{Y}_i \leftarrow Y_i + \xi_i$                                  ▷ Noise regularization

8:         $\tilde{Y}_{\text{aug},i} \sim N(g^a(\tilde{Y}_i), \varepsilon^2)$                          ▷ Variational augmentations

9:         $\mathcal{L}_{\text{NLL}}(\beta_a, \theta_a) \leftarrow -\frac{1}{n} \sum_{i=1}^b \log \hat{\mathbb{P}}_{\beta_a, \theta_a}(Y = \tilde{Y}_i \mid a)$   ▷ (1) Negative log-likelihood loss

10:         $\{(U_{Y^1,j}, U_{Y^2,j})\}_{j=1}^b \sim \text{Unif}(0,1)^2$

11:         $\{(\hat{Y}_{\text{aug},a,j}, \hat{Y}_{a,j})\}_{j=1}^b \leftarrow \hat{F}_a\left(\{(U_{Y^1,j}, U_{Y^2,j})\}_{j=1}^b\right)$

12:         $\mathcal{L}_{\mathbb{W}}(\beta_a) \leftarrow \int_0^1 |\hat{\mathbb{F}}_{\hat{Y}_a}^{-1}(q) - \hat{\mathbb{F}}_{Y_a}^{-1}(q)| \, dq$                      ▷ (2) Wasserstein loss

13:         $\mathcal{L} \leftarrow \mathcal{L}_{\text{NLL}}(\beta_a, \theta_a) + \mathcal{L}_{\mathbb{W}}(\beta_a)$

14:         optimization step for $\beta_a, \theta_a$ wrt. $\mathcal{L}$ with learning rate $\eta$

15:     **end for**

16: **end for**

17: **Output.** Pretrained flows parameters: residual NFs $(\beta_0, \beta_1)$ and variational augmentations $(\theta_0, \theta_1)$

18:

19: $\overline{\beta}_a, \underline{\beta}_a \leftarrow \beta_a; \quad \overline{\theta}_a, \underline{\theta}_a \leftarrow \theta_a;$                      ▷ Copying counterfactual flow parameters

20: $[\hat{l}_a, \hat{u}_a] \leftarrow [\min_i(Y_i); \max_i(Y_i)]$

21: **for** $k = 0$ to $n_Q + n_{CQ}$ **do**                             ▷ **Query & curvature-query stages**

22:     **for** $(\beta_a, \theta_a) \in \{(\overline{\beta}_a, \overline{\theta}_a), (\underline{\beta}_a, \underline{\theta}_a)\}$ **do**

23:         $\{A_i = a, Y_i\}_{i=1}^b \leftarrow$ minibatch of size $b$ from $\mathcal{D}_a$

24:         same procedure as in lines 7–12 to evaluate $\mathcal{L}_{\text{NLL}}(\beta_a, \theta_a)$ and $\mathcal{L}_{\mathbb{W}}(\beta_a)$

25:         $\mathcal{L} \leftarrow \mathcal{L}_{\text{NLL}}(\beta_a, \theta_a) + \mathcal{L}_{\mathbb{W}}(\beta_a)$

26:         $(y', \{Y'_{\text{aug},j}\}_{j=1}^b \sim N(g^{a'}(y'), \varepsilon^2))$                 ▷ Variational augmentations

27:         $\{(\tilde{u}_{Y^1,j}, \tilde{u}_{Y^2,j})\}_{j=1}^b \leftarrow \hat{F}_{a'}^{-1}\left(\{(y', Y'_{\text{aug},j})\}_{j=1}^b\right)$

28:         $\{(\hat{Y}_{\text{aug},j}, \hat{Y}_j)\}_{j=1}^b \leftarrow \hat{F}_a\left(\{(\tilde{u}_{Y^1,j}, \tilde{u}_{Y^2,j})\}_{j=1}^b\right)$

29:         $\hat{Q}_{a' \to a}(y') = \frac{1}{b} \sum_{j=1}^b \hat{Y}_j$                           ▷ Estimated ECOU [ECOT]

30:         **if** $\hat{Q}_{a' \to a}(y') \notin [\hat{l}_a, \hat{u}_a]$ **and** $k < n_Q$ **then**

31:             **continue**                              ▷ Non-informative bounds

32:         **else**

33:             $\mathcal{L}_Q(\beta_{a'}, \theta_{a'}, \beta_a) \leftarrow \text{Softplus}(\mp \hat{Q}_{a' \to a}(y'))$      ▷ (3) Counterfactual query loss

34:             $\mathcal{L} \leftarrow \mathcal{L} + \lambda_Q \mathcal{L}_Q(\beta_{a'}, \theta_{a'}, \beta_a)$

35:         **end if**

36:         **if** $k \geq n_Q$ **then**

37:             $\hat{y} \leftarrow \hat{Q}_{a' \to a}(y'); \quad (\hat{y}, \{\hat{Y}_{\text{aug},j}\}_{j=1}^b \sim N(g^a(\hat{y}), \varepsilon^2))$    ▷ Variational augmentations

38:             $\{(\hat{u}_{Y^1,j}, \hat{u}_{Y^2,j})\}_{j=1}^b \leftarrow \hat{F}_a^{-1}\left(\{(\hat{y}, \hat{Y}_{\text{aug},j})\}_{j=1}^b\right)$

39:             $\mathcal{L}_\kappa(\beta_a, \theta_a) \leftarrow \frac{1}{b} \sum_{j=1}^b \kappa_1(\hat{u}_{Y^1,j}, \hat{u}_{Y^2,j})$              ▷ (4) Curvature loss

40:             $\mathcal{L} \leftarrow \mathcal{L} + \lambda_\kappa \mathcal{L}_\kappa(\beta_a, \theta_a)$

41:         **end if**

42:         optimization step for $\beta_a, \theta_a$ wrt. $\mathcal{L}$ with learning rate $\eta$

43:         EMA update of the flow parameters $(\beta_a, \theta_a)$

44:     **end for**

45: **end for**

46: **Output.** $\hat{Q}_{a' \to a}(y')$ evaluated with smoothed with EMA $(\overline{\beta}_a, \overline{\theta}_a)$ and $(\underline{\beta}_a, \underline{\theta}_a)$

---

# H Synthetic experiments

## H.1 Datasets

We conduct the experiments with datasets, drawn from two synthetic datasets. In the first dataset, we use

$$\begin{cases} Y \mid 0 \sim \mathbb{P}(Y \mid 0) = N(0,1), \\ Y \mid 1 \sim \mathbb{P}(Y \mid 1) = N(0,1), \end{cases} \tag{105}$$

and, in the second,

$$\begin{cases} Y \mid 0 \sim \mathbb{P}(Y \mid 0) = \text{Mixture}(0.7\,N(-0.5, 1.5^2) + 0.3\,N(1.5, 0.5^2)), \\ Y \mid 1 \sim \mathbb{P}(Y \mid 1) = \text{Mixture}(0.3\,N(-2.5, 0.35^2) + 0.4\,N(0.5, 0.75^2) + 0.3\,N(2.0, 0.5^2)). \end{cases} \tag{106}$$

Importantly, the ground truth SCMs (and their $\kappa$) for both datasets remain unknown, and we only have access to the observational distributions. This is consistent with the task of partial counterfactual identification.

## H.2 Additional results

In Fig. 14, we provide additional results for the synthetic experiments. For our APID, we set $\lambda_Q = 1.0$ and vary $\lambda_\kappa \in \{0.0, 0.5, 1.0, 5.0\}$. Here, we see that estimated by APID bounds are located closer to (i) the theoretic upper and lower bounds of BGMs and (ii) closer to each other.

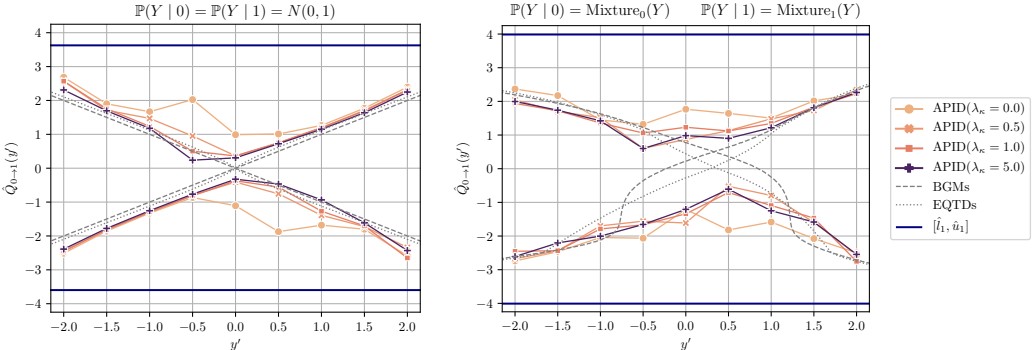

Figure 14: Additional results for partial counterfactual identification of the ECOU across two datasets. Reported: theoretical bounds of BGMs and mean bounds of APID over five runs. Here, $\lambda_Q = 1.0$.

## H.3 Runtime

Table 2 provides the duration of one training iteration of our APID for different training stages. Namely, we report the mean and std. across all the experiments (total of 720 runs) for the burn-in, the query, and the curvature-query stages. In our experiments, the long training times are attributed to the reverse transformations in the residual normalizing flows (as they require fixed point iterations) and to the computation of the Hessian to evaluate the curvatures.

| Training stage | Number of training iterations | Evaluated losses | Duration per training iteration (in s) |
|---|---|---|---|
| Burn-in stage | $n_B = 500$ | $\mathcal{L}_{\text{NLL}}, \mathcal{L}_{\mathbb{W}}$ | $2.60 \pm 1.72$ |
| Query stage | $n_Q = 100$ | $\mathcal{L}_{\text{NLL}}, \mathcal{L}_{\mathbb{W}}, \mathcal{L}_Q$ | $11.04 \pm 10.65$ |
| Curvature-query stage | $n_{CQ} = 500$ | $\mathcal{L}_{\text{NLL}}, \mathcal{L}_{\mathbb{W}}, \mathcal{L}_Q, \mathcal{L}_\kappa$ | $39.27 \pm 27.26$ |

Table 2: Total runtime (in seconds) for different stages of training for our APID. Reported: mean $\pm$ standard deviation across all the experiments, which amount to a total of 720 runs (lower is better). Experiments were carried out on 2 GPUs (NVIDIA A100-PCIE-40GB) with IntelXeon Silver 4316 CPUs @ 2.30GHz.

# I Case study with real-world data

In our case study, we want to analyze retrospective "what if" questions from the COVID-19 pandemic. Specifically, we ask what the impact on COVID-19 cases in Sweden would have been, had Sweden implemented a stay-home order (which Sweden did not do but which is under scrutiny due to post-hoc legal controversies).

## I.1 Dataset

We use multi-country data from [7].[9] It contains weekly-averaged number of recorded COVID-19 for 20 Western countries. The outcome, $Y$, is the (relative) case growth per week (in log), namely, the ratio between new cases and cumulative cases. The treatment, $A \in \{0, 1\}$, is a binary variable referring to either no lockdown or enforcement of a stringent lockdown, respectively. We assume the data is i.i.d. and comes from a randomized control trial, i.e., there is no confounding between $Y$ and $A$. We filtered out the weeks where the cumulative number of cases is smaller than $50$, which resulted in $n = n_0 + n_1 = 136 + 124$ observations in total, for both treatments, respectively. We plot a histogram of case growth per week depending on the treatment in Fig. 15, i.e., empirical densities of distributions $\hat{\mathbb{P}}(Y \mid A = 0)$ and $\hat{\mathbb{P}}(Y \mid A = 1)$.

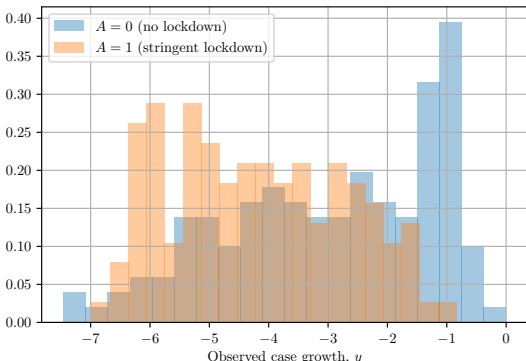

Figure 15: Empirical distributions of case growth per week (in log) for both treatments. Larger values mean a larger incidence.

We assume that the observational distributions $\mathbb{P}(Y \mid a)$ are induced by some (unknown) SCM of class $\mathfrak{B}(C^2, 2)$. Then, we fit our APID the same way as in synthetic experiments (see Appendix G). We additionally report the BGMs-EQTDs identification gap based on empirical CDF of the observational distributions.

## I.2 Results

Fig. 16 shows the results for (upper) partial counterfactual identification of the ECOU for our method. Hence, the ECOU reports for the expected counterfactual case growth for countries without stay-home orders, i.e., the weekly case growth (in log) if a stay-home order would have actually been enforced. We also highlight the upper bounds on the ECOU for Sweden during weeks 10–13 of 2020, where no lockdown was implemented. Evidently, the ECOU would lie below the decision boundary for BGMs-EQTDs and for certain levels of $\lambda_\kappa$. This implies that Sweden could have had significantly lower case growth had it implemented a stay-home order, even when allowing for some level of non-linearity between outcome and independent latent noises that interact with the treatment, e.g., cultural distinctions, level of trust in government, etc. The results of the COVID-19 case study demonstrate that our CSM and its instantiation with APID could be applied for decision-making with real-world data.

---

[9]The data is available at `https://github.com/nbanho/npi_effectiveness_first_wave/blob/master/data/data_preprocessed.csv`.

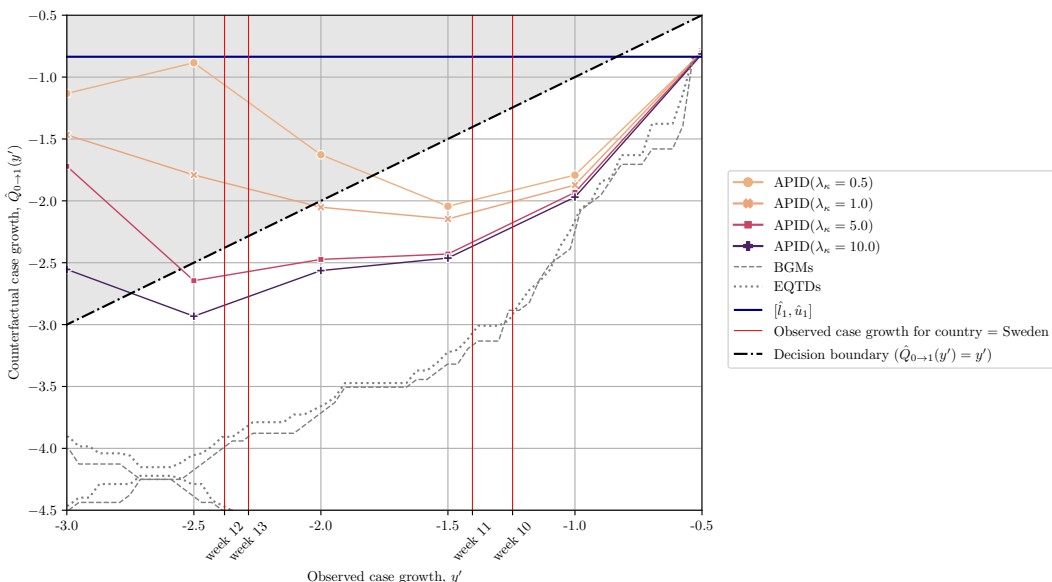

Figure 16: Case study where we analyze retrospective "what if" questions from the COVID-19 pandemic. Reported: BGMs-EQTDs identification gap and ECOU upper bounds of APID over five runs (in log). The gray-shaded area above denotes the decision boundary with the negative counterfactual effect, namely where a stay-home order would have led to an increase in case growth, respectively. The red lines show the observed case growth for Sweden during weeks 10–13 of 2020, where no stay-home order was implemented.

