# OpenReview forum: "Partial Counterfactual Identification of Continuous Outcomes with a Curvature Sensitivity Model"
_NeurIPS.cc/2023/Conference — NeurIPS 2023 spotlight_

### Official Review · Reviewer_23yB · 2023-07-06

**Soundness:** 3 good
**Presentation:** 2 fair
**Contribution:** 3 good
**Rating:** 6
**Confidence:** 3

**Summary:**

This paper studies the partial identification of counterfactual probabilities from observational data. More specifically, the authors consider the standard bandit model containing a treatment A and a reward Y; exogenous variables U exist affecting both A and Y. The treatment A is a binary variable; while the reward Y could take any arbitrary real value bounded in an interval [l, u]. The goal is to evaluate the counterfactual probability P(y_a | a’, y’) from the observational data drawn from P(Y, A). The author first shows that this learning setting is ill-posed without any additional assumptions. That is, the learner could only obtain non-informative bound [l, u] from the observational data. This observation suggests that one should explore additional parametric assumptions to obtain informative bounds. More specifically, the authors propose a sensitivity model called Curvature Sensitivity Model. They further propose an implementation of our Curvature Sensitivity Model in the form of a novel deep generative model, called Augmented PseudoInvertible Decoder.

**Strengths:**

- This paper is well written and well organized. The authors provide extensive examples and graphical illustrations. This is much appreciated.

- Bounding counterfactual probabilities over a continuous outcome from the observational data is a challenging problem in causal inference. The authors study partial identification in multi-armed bandit models, which are widely applied in reinforcement learning literature. The target query P(y_a | a’, y’) is called the expected counterfactual outcome of un]treated ECOU; its applications include fairness analysis in machine learning (Kushner et al., 2018). Since ECOU is generally non-identifiable and existing bounding algorithms focus on discrete domains, this paper could have an impact on the causal inference community.

- The non-informative bounding result in Theorem 1 is interesting. As far as I am aware, this result is novel. The authors also provide a novel sensitivity model of Curvature Sensitivity Model (CSM), which permits one to obtain informative bounds by bounding the curvature of level sets of the functions. They further show that CSMs generalize existing parametric assumptions for point counterfactual identification methods when the bound of the curvature is set to zero.

**Weaknesses:**

- This paper studies the bounding of a specific counterfactual probability, called ECOU in canonical bandit models. However, some statements in the abstract sound more general than it is. For instance, the paper states "We prove that, in general, the ignorance interval of the counterfactual queries has non-informative bounds, already when functions of structural causal models are continuously differentiable." Based on my reading, the non-informative bounds in Theorem 1 only apply to bounding ECOU in bandit models. However, if we consider bounding a causal effect P(y_a) in the bandit model, Manski's bound (1990) has shown to be informative and applicable to continuous reward signals.

- The simulation plots could be improved. For instance, it is unclear what the ground truth ECOUs are in Figure 7. Without that, it is hard to evaluate the validity of the proposed bounding strategy.



**Questions:**

- What is the actual ECOU in Figure 7?

**Limitations:**

Assumptions behind the proposed algorithms are explicitly stated. This paper is theoretical and its long-term societal impact is not immediate to see.

---

> ### Author Rebuttal · Authors · 2023-08-09
>
> We are grateful for the review. It is great to hear that you found our paper well-written and well-organized, and that you think that the paper will have an impact on the causal inference community. We would like to address mentioned weaknesses and open questions.
>
> ### Response to weaknesses
> We want to stress, that our setting is different from (causal) bandits in reinforcement learning, as we do **not** have access to the environment. As such, we do **not** have the ability to perform experiments and, therefore, do not aim to maximize some reward. Instead, we have a setting based on observational/experimental data (also called logged data) with treatment and outcome (e.g., from a randomized controlled trial), and we aim to infer the expectation of the counterfactual outcome.
>
> The Manski's bound (1990) for the causal effect P(y_a) is fundamentally **different** from our counterfactual query (see examples in [1]), namely ECOU[ECOT]. As such, it is **not** applicable to our setting. The query P(y_a) lies only at the interventional (second) layer of Pearl’s hierarchy of causation [2] (see Table 1 in Appendix A.1). Generally, the inference on different layers almost never coincides (this is known as the *Causal Hierarchy Theorem* [2]).
>
> In our setting, ground-truth bounds would be desirable but are *intractable*. The reason is the following: For a given value of $\kappa$, it is computationally infeasible to derive ground truth bounds for ECOU [ECOT], even when the ground truth observation density is known. The ground-truth bounds would require solving a constrained optimization task including partial derivatives and Hausdorff integration. This is intractable, even for such simple distributions as standard normal.
>
> ### Response to questions
> Unfortunately, while ground-truth bounds are desirable, they are *intractable*. As we stated above, the reason is the following. For a given value of $\kappa$, it is computationally infeasible to derive ground truth bounds for ECOU [ECOT], even when the ground truth observation density is known. The ground-truth bounds would require solving a constrained optimization task including partial derivatives and Hausdorff integration. This is *intractable*, even for such simple distributions as standard normal.
>
> References:
> - [1] Alexander Balke and Judea Pearl. “Counterfactual probabilities: computational methods, bounds and applications”. In: Uncertainty Proceedings. Elsevier, 1994, pp. 46–54.
> - [2] Elias Bareinboim et al. “On Pearl’s hierarchy and the foundations of causal inference”. In: Probabilistic and Causal Inference: The Works of Judea Pearl. Association for Computing Machinery, 2022, pp. 507–556.

---

### Official Review · Reviewer_53D4 · 2023-07-06

**Soundness:** 3 good
**Presentation:** 4 excellent
**Contribution:** 4 excellent
**Rating:** 7
**Confidence:** 4

**Summary:**

The authors study the problem of counterfactual identification of continuous outcome variable with a specific causal graph. They show that the expected counterfactual outcome of (un)treated has non-informative bound when the function class is arbitrary smooth. Then, they introduce a new model called CSM that give informative bounds. Finally, they propose a deep generative based method for partial counterfactual inference in CSM.

**Strengths:**

They study an important problem. The paper is well-written and the Appendix is educational.

The analysis is quite interesting as it combines differntial geometry and causal inference. This opens a new door to improve the current results in causal inference and likely to develop a general framework for counterfactual analysis and more.

The presented examples in both the main text and the appendix are informative.

Theoretical results are quite interesting. In particular, Theorem 1 and 2.

**Weaknesses:**

The main question is about the generalizability of this method to more complex causal grpahs. It is not hard to see that as the causal diagram becomes a bit more complex, then applying similar analysis might be quite challenging.

It is a bit confusing that CSMs with arbitrary $\kappa$ lead to informative bounds, i.e., $l(\kappa,d)>l_a$ while, for instance, $\mathcal{B}(C^2,d)$ is not informative. Because, as it is also mentioned in the text, as $\kappa\rightarrow\infty$, CSM$(\kappa)\rightarrow \mathcal{B}(C^2,d)$. What is the interpretaion of this?

The proof of Theorem 2 is of existance type that is the form of the informative bounds and how they depend on $\kappa$ and $d$ are not given. But is it still possible to say something about their dependencies, e.g., $l(\kappa,d)$ is decreasing with respect to $\kappa$?

Although, the augmented Pseudo-Invertible decoder method is a fine application of deep generative methods but it is not clear how sensitive it is with respect to the model hyperparametersa and sample size.


**Questions:**

See above

**Limitations:**

 The authors addressed the limitations.

---

> ### Author Rebuttal · Authors · 2023-08-09
>
> Thank you for your review! We appreciate that you found our paper interesting, important, and well-written. We would like to respond to the mentioned weaknesses.
>
> - Regarding the generalizability of our method to the more complex graphs: Our CSM generalizes in theory to other Markovian-SCMs (see discussion in Appendix E). This applies to settings where we observe other parents of the outcome, independent of the latent noise.
>
>   We would like to stress that, to the best of our knowledge, our paper is the first to address partial counterfactual inference for continuous outcomes (therefore we considered a rather simple causal diagram). For this, we are first to provide a measure-theoretic formulation aimed at the partial identification task. Further, we are first to formulate the results of non-informative bounds for smooth functions. We think that our theoretical results are an essential foundation for future work aimed at studying more general causal graphs.
>
> - Our CSM with $\kappa = \infty$ (or equivalently $\lambda_\kappa = 0$) should lead to non-informative bounds. However, the heuristic implementation, i.e., APID, sometimes encounters computational and optimization issues that may lead to informative bounds (e.g., in Figure 13 in Appendix H). For example, the gradual “bending” of the level sets is sometimes numerically unstable during training, and some runs omit “bending” at all as it would require passing through the high loss value region. Therefore, the computed bounds for ECOU [ECOT] may be too tight.
> As reviewer 8Zs8 nicely summarized, “In an ideal world, the computational method (i.e., the Augmented Pseudo-Invertible Decoder in Section 5) would take a given value of $\kappa$ and return the [exact] bounds as defined in Definition 2. However, somewhat understandably, returning the exact bounds is not necessarily numerically feasible.”  Instead, our AIPD offers numerical estimates for them.
>
>   **Action**: We will expand the discussion of our experiments and the above limitations.
>
> - By construction, it is guaranteed that, with decreasing $\kappa$, the bounds for ECOU [ECOT] will shrink, as we decrease the class of permissible functions in the constraints of the optimization problem (Definition 2).
>
>   **Action**: We will add a formal statement as a new Corollary after Theorem 2.
>
> - Our APID should be seen as a proof of concept rather than a fully-fledged out-of-the-box method. Nevertheless, we identified a specific set of hyperparameters (incl. hidden dimensionalities, regularization terms, etc.) to work well in all the experiments  (see Appendix G.2), including the newly added real-world study. In particular, we found that the goodness-of-fit on the held-out subset (e.g., test log-likelihood) was consistently favorable after the burn-in stage of training for all the experiments.
>
>   Regarding the sample size, our APID works well as soon as it can fit the univariate density of the outcome well. In our newly added case study, the total number of observations was only $n = n_0 + n_1 = 136 + 124$, for which still obtained robust results.

---

### Official Review · Reviewer_JMz8 · 2023-07-07

**Soundness:** 4 excellent
**Presentation:** 3 good
**Contribution:** 4 excellent
**Rating:** 7
**Confidence:** 4

**Summary:**

This paper studies the partial identifiability of counterfactual queries and proposes an interesting proof of general non-identifiability resulting in non-informative bounds. The authors build upon the intuition used for this proof to propose a new class of structural causal model using principal curvatures. Under this assumption, the authors can achieve informative partial counterfactual identification. The authors then propose an architecture that leverages this assumption.

**Strengths:**

This paper proposes a systematic and rigorous study of the partial counterfactual estimation problem. This is exemplified by the breadth and depth of literature search and by the organization of the related works. While general non-identifiabilty of counterfactuals is known, theorem 1 shows that constraints on the smoothness of repsonse function does not improve the identifiability, which I think is novel.

The proposed curvature sensitivity model, motivated by Theorem 1, is shown to benefit from the identifiability properties which are theoretically demonstrated. This causal model structure seems to be fairly general and appears to generalize concepts such as non-intersection of response surfaces (which is pointed at in the experiments).

The authors further show that the theoretical properties of the CSM can be leveraged empirically be proposing a new architecture.

**Weaknesses:**

While there is significant effort in bringing intuition regarding how to construct counterfactuals (bending of the counterfactual level sets), I believe more intuition could be given regarding the general applicability of the CSM in practice. It is unclear when it seems reasonable to assume this model would be valid, and hence the partial counterfactual estimation correct. I appreciate that this is a general problem in counterfactual estimation but more insight into the applicability would be welcome.

While there is a strong theoretical work behind this paper, it is more frugal in terms of experiments. I would encourage the authors to provide more evaluations of their approach, with different types of repsonse surfaces and noise levels. The design of more experiments could potentially lead to more intuition in the assumptions of the method.

**Questions:**

Could the author motivate the CSM assumption from a more practical perspective ?

Extending the experiments section would help strenghten the paper.

**Limitations:**

Limitations and assumptions are clearly stated.

---

> ### Author Rebuttal · Authors · 2023-08-09
>
> Thank you for the positive evaluation of our work. We are happy that you consider our paper a novel, systematic, and rigorous study. We would like to discuss the mentioned weaknesses and questions in the following.
>
> - We agree that the $\kappa$ in our CSM may be perceived — in parts — as an abstract sensitivity parameter. Nevertheless, the sensitivity parameter $\kappa$ lends to a natural interpretation. It can be interpreted as the *level of non-linearity between the outcome and its latent noises that interact with treatment*. To illustrate this, let us consider two following scenarios:
>    - (i) When the treatment does not interact with any of the latent noise variables, then we can assume w.l.o.g. a BGM (thus: deterministic counterfactual outcome). There is no loss of generality, as all the other SCMs with $d > 1$ are equivalent to this BGM (i.e., their level set bundles coincide for both factual and counterfactual transformation).
>    - (ii) When the treatment interacts with some latent noise variables, then the counterfactual outcome becomes random, and we cannot assume a BGM but we have to use our CSM. In this case, $d$ corresponds to the number of latent noise variables which interact with the treatment. Hence, $\kappa$ bounds the level of non-linearity between the outcome and the noise variables. More formally, $\kappa$ and principal curvatures can be seen as the largest coefficient of the second-order term in the Taylor expansion of the level set (see Eq. 15 in the Appendix). This interpretation of the $\kappa$ goes along with human intuition (see [1]): when we try to imagine counterfactual outcomes, we tend to “traverse” all the possible scenarios which could lead to a certain value. If we allow for highly non-linear scenarios, which interact with treatment, we also allow for more extreme counterfactuals, e.g., interactions between treatment and rare genetic conditions.
>
>   Importantly, in order to verify the particular value of $\kappa$, we would have to measure all the latent noises which interact with treatment. Verifying counterfactual inference is a very complex task, and can be achieved by, e.g., combining several datasets (known as causal marginal problem [2], yet which has been done only for discrete variables and not for continuous variables as in our work). Still, we are confident that our paper provides a valuable theoretical foundation for future research and can help practical applications (see next bullet item).
>
>   **Action**: We will add the interpretation of $\kappa$ and its intuition to our revised paper.
>
> - We experimented with different coefficients for the $\lambda_\kappa$ (which correspond to different bounds on the curvature of level sets). Our CSM considers *all the possible response surfaces* with a certain max curvature of the level sets, even when the dimensionality of the latent noise is set to $d=2$ (see Corollary 4 in Appendix D).
>
>   To address the issue of applicability and to strengthen the intuition behind our CSM, we performed an additional experiment with a real-world case study (see **PDF**). Therein, we adopt our CSM to answer “what if” questions related to whether the lockdown was effective during the COVID-19 pandemic. Specifically, we ask what the impact on COVID-19 cases in Sweden would have been, had Sweden implemented a stay-home order (which Sweden did not do but which is under scrutiny due to post-hoc legal controversies). Our results imply that Sweden could have had significantly lower cases had it implemented a stay-home order, even when allowing for some level of non-linearity between outcome and independent latent noise variables that interact with the treatment, e.g., cultural differences, level of trust in government, etc.
>
>   **Action**: We will add our case study with real-world data to our revised paper.
>
> References:
> - [1] Celar, Lenart, and Ruth MJ Byrne. "How people reason with counterfactual and causal explanations for Artificial Intelligence decisions in familiar and unfamiliar domains." Memory & Cognition (2023): 1-16.
> - [2] Luigi Gresele et al. “Causal inference through the structural causal marginal problem”. In: International Conference on Machine Learning. 2022.

---

### Official Review · Reviewer_8Zs8 · 2023-07-27

**Soundness:** 3 good
**Presentation:** 3 good
**Contribution:** 2 fair
**Rating:** 6
**Confidence:** 3

**Summary:**

This paper presents a method for obtaining bounds on continuous counterfactual outcomes.  In general, level-3 counterfactual outcomes (what would have happened under action $a'$, given knowledge of what did happen under action $a$) are not identifiable from either interventional or observational data.  Prior work makes structural assumptions on the underlying structural causal model (e.g., monotonicity with respect to a single unobserved noise variable) to obtain identification, while this work instead makes an assumption on the principal curvature of level sets with respect to two unobserved noise variables.  A computational approach is provided to obtain bounds based on this principal curvature assumption, and is demonstrated on two simple synthetic cases.


**Strengths:**

In terms of originality and clarity, this paper presents a novel approach to obtaining bounds on continuous counterfactual outcomes, and the contribution is clearly placed in the context of related work.  Having seen a few papers on the general topic of partial identification, I found Section A (Extended Related Work) in the Appendix to be a well-written, well-researched, and generally helpful overview of how the current work differs from prior approaches, primarily in the problem considered.

Moreover, in terms of clarity and quality, I found Example 1 (Figure 3) to be a very helpful example of counterfactual non-identifiability, and a good explanation of the intuition for how "bending" the counterfactual level sets around the factual level set leads to a different distribution of counterfactual outcomes.


**Weaknesses:**

## Summary
I see two major weaknesses in this paper, and look forward to discussion with the authors during the response period if I've misunderstood anything below.  Both weaknesses relate to the significance of the approach.

1. How should domain experts choose the parameter $\kappa$?  Choosing the right bound on the principal curvature of level sets doesn't seem like something a domain expert would plausibly be able to do.
2. Even if we knew the right $\kappa$, can we guarantee that the computational approach will give bounds that are at least conservative (e.g., outer bounds on the true bounds implied by $\kappa$) if not tight?

Overall, my current score incorporates my view that these two weaknesses are genuinely challenging problems, so I am somewhat sympathetic to the paper in its current state, but I would at least like to see more discussion of these challenges in the main paper.

I'll discuss these weaknesses in more detail below.  I am basing my score primarily on these points, though I also had an important clarifying question regarding the ability of the approach to "cover the entire identifiability spectrum".

## Details of major concerns

(1) **Domain Knowledge for $\kappa$?** One of the appealing properties of the given approach is that it can interpolate between the extremes of identifiability (though see my clarifying question (3) below).  However, there are many trivial ways to interpolate between e.g., point-identification and the non-informative bounds (e.g., by simply choosing a point between them). The value of sensitivity analysis approaches (in my view) lies in the ability to translate domain knowledge into informative bounds.

In other areas of sensitivity analysis (for e.g., interventional queries in the presence of unobserved confounding) that rely on domain knowledge to derive bounds, there is an emphasis on creating additional tools to help end-users calibrate their choices of these hyperparameters by e.g., benchmarking (see Figure 2 of Cinelli and Hazlett 2019, full citation below).  Informally, this type of benchmarking allows one to say "an unobserved confounder would have to be 3x as strong as the variable 'age' in order to change our conclusions", a statement that is a bit easier for domain experts to engage with versus reasoning about the value of an abstract hyperparameter.

I'm aware that this example deals with a different type of causal query (i.e., counterfactuals vs interventional queries), and that the lack of identifiability in the current paper is fundamentally different from unobserved confounding - my point is that for any partial identification approach that derives informative bounds via "domain knowledge", like the current approach, a bit more effort is required to make it plausible for domain experts to convert their domain knowledge into the required (abstract) hyperparameters, like bounding the principal curvature of counterfactual level sets.

I would be interested in any comments from the authors on this point.

Citation: Carlos Cinelli , Chad Hazlett, Making Sense of Sensitivity: Extending Omitted Variable Bias, Journal of the Royal Statistical Society Series B: Statistical Methodology, Volume 82, Issue 1, February 2020, Pages 39–67, https://doi.org/10.1111/rssb.12348

(2) **Corresponding between computational bounds and theoretical bounds**:  Supposing we know the right value of $\kappa$, Definition 2 defines an upper and lower bound on the quantity of interest (e.g., the ECOU), with the additional constraint (e.g., in Theorem 2) that all SCMs satisfy Assumption $\kappa$ and are in the class $\mathcal{B}(C^2, d)$.  For the practical algorithm, we use $d = 2$.

In an ideal world, the computational method (i.e., the Augmented Pseudo-Invertible Decoder in Section 5) would take a given value of $\kappa$ and return the bounds as defined in Definition 2.  However, somewhat understandably, returning the exact bounds is not necessarily feasible.

Instead, we are left with heuristics - we optimize an objective over a certain set of SCMs where one of the four terms relates to maximizing / minimizing the counterfactual outcome, and is controlled by the hyperparameter $\lambda_Q$.  It was not clear to me how this term trades off with the other terms, e.g., the extent to which we can view the optimization problem as choosing a fixed constraint on $\kappa$ and then maximizing / minimizing the counterfactual subject to this constraint among all observationally-equivalent models.

**The use of practical heuristics would be less concerning if this approach resulted in less informative bounds than those in Definition 2, but as I understand it, it may result in *more* informative bounds.**

To clarify, this point is briefly discussed in Appendix E (discussion of extensions and limitations) under "Tight bounds". However, my concern is not that the bounds are insufficiently tight (e.g., too wide and could be narrowed), but that they might be *too narrow* (e.g., the theoretical bounds are actually wider than the bounds produced by the algorithm).

I believe there is some evidence to suggest that these heuristics change the interpretation of the results.  For instance, I would expect that as $\lambda_{\kappa}$ changes, the bound should monotonically increase/decrease. However, looking closely at Figure 7 suggests that in practice there is some crossover (see e.g., $y' = -0.5$, where the lower bound for $\lambda_{\kappa} = 1$ goes above the line for $\lambda_{\kappa} = 10$).  I believe my observation below relating to Figure 13 in the appendix (where $\lambda_{\kappa} = 0$ still gives informative bounds) is also related to this point.

Overall, it would be good to see a comparison, perhaps for a very simple example, between the ground-truth bounds in Definition 2 and the bounds returned by the approach.

(3) **Clarification regarding non-informative bounds with $\kappa=\infty$**: It is stated in Corollary 4 that by setting $d = 2$ and varying $\kappa$, we can "cover the entire identifiability spectrum".  I took this statement to mean that if e.g., there is no restriction on $\kappa$, then the resulting bounds should be non-informative (i.e., equal to $[l_a, u_a]$), and that for $\kappa = 0$, all values of $d$ yield the same point-identified solution.  I'm deriving my understanding primarily from the explanation given in the introduction (see lines 72-73).  Is my understanding correct?

If my understanding is correct, then what is going on in Figure 13 in the appendix?  It appears that setting $\lambda_{\kappa} = 0$ (corresponding to no constraints on $\kappa$) does not result in non-informative bounds.  Is this due to the (perhaps necessary) mismatch between the computational approach and perfectly solving the optimization in Definition 2?

## Minor Points

As a minor point, it may be worth clarifying where we are interested in taking a maximum/minimum (e.g., over observationally-equivalent SCMs) and where we are interested in taking an average (e.g., over the counterfactual posterior for a single SCM).  E.g., when I first read the procedure for abduction on lines 341-344, I was concerned that the approach draws samples from the pre-image, as opposed to looking for $U_1, U_2$ which yield the maximum/minimum counterfactual outcomes, before I realized that this approach makes sense for a fixed SCM to compute a counterfactual expectation like ECOU / ECOT.

I also found Figure 6 difficult to follow, especially the bi-directional arrows for "connections".


**Questions:**

1. How would you go about helping domain experts choose an appropriate value of $\kappa$ or $\lambda_{\kappa}$ in practice?
2. Is there any guarantee that the bounds derived from this approach will be less informative than the theoretical bounds?  Or, should we generally expect the bounds to be more informative, since we will inevitably fail to find the SCMs that yield the true maximum / minimum for a given $\kappa$?
3. Related to the previous question, is my understanding of "cover the entire identifiability spectrum" correct in Corollary 4?  If so, it may be worth revising the theoretical statement to make this clear.  If my understanding is correct, then what is going on in Figure 13, i.e., why does setting $\lambda_{\kappa} = 0$ still yield informative bounds?


**Limitations:**

In the main paper, I would have liked to see more discussion of the limitations outlined in the "weaknesses" section above.

---

> ### Author Rebuttal · Authors · 2023-08-09
>
> We are very grateful for such a rigorous, helpful, and in-depth review. We also appreciate that the reviewer found our paper clearly written, our method novel, and our contribution well-placed in the related work. Here is our answer to the questions:
>  - (1) We stress that counterfactual inference is a non-trivial problem in statistics and probability. Even in human cognition, counterfactuals are much more complicated than causal statements [1]. To the best of our knowledge, our paper is the first to address partial counterfactual inference for continuous outcomes. For this, we provided a measure-theoretic formulation and formulated the results of non-informative bounds for smooth functions. Our CSM is one viable solution (out of many) to build a sensitivity model. We agree that κ in our CSM may be perceived as an abstract sensitivity parameter. Yet, it can be interpreted as the *level of non-linearity between the outcome and its latent noises that interact with treatment*. To illustrate this, let us consider two following cases:
>    - (i) When the treatment does not interact with any of the latent noises, then we can assume w.l.o.g. a BGM. There is no loss of generality, as all the other SCMs with d > 1 are equivalent to this BGM (see Ex. 6).
>    - (ii) When the treatment interacts with some latent noise variables we cannot assume a BGM and have to use our CSM. In this case, d corresponds to the number of latent noise variables which interact with the treatment. Hence, κ bounds the level of non-linearity between the outcome and the noise variable. More formally, κ and principal curvatures can be seen as the largest coefficient of the second-order term in the Taylor expansion of the level set (see Eq. 15 in App. B). This interpretation of the κ goes along with human intuition (see [1]): when we try to imagine counterfactual outcomes, we tend to “traverse” all the possible scenarios which could lead to a certain value. If we allow for highly non-linear scenarios, we also allow for more extreme counterfactuals, e.g., interactions between treatment and rare genetic conditions.
>
>    **Action**: We will add the interpretation of κ and the intuition behind its choice to our revised paper. To strengthen the practical value of our CSM, we provided a new case study (see **PDF**). Therein, we adopt our CSM to answer “what if” questions related to whether the lockdown was effective during the COVID-19 pandemic.
>
> - (2) We agree that, for a given value of κ, it is computationally infeasible to derive ground-truth (GT) bounds, even when the GT observation density is known. This is different from, e.g., layer 2 models like MSM, where the GT solution could be expressed in terms of conditional quantiles. The GT bounds in Def. 2 require solving a constrained optimization task including partial derivatives. This is intractable, even for simple distributions as normal.
>
>   Thus, we resort to using tractable alternatives and, hence, design our APID. The approach of turning constraints into parts of the loss is a common practice in deep learning for partial identification [2, 3]. In our case, this involves either adding the curvature loss controlled by λ_κ and the query loss controlled by λ_Q. The exact relationship between κ and corresponding λ_κ is unknown, but they are inversely connected. In general, this relationship is usually known only for very simple models, like ridge or lasso regressions. Regarding the choice of λ_Q, we provided additional experiments in our original supplements in App. H.2. Therein, the bounds are simply moved towards/away from BGMs-EQTDs bounds, but their relative location almost does not change. We acknowledge that our APID is a proof of concept rather than a fully-fledged method. Hence, for the multi-modal distribution in Fig. 7, APID sometimes yielded inaccurate bounds (e.g., too tight), mainly due to the computational instability (i.e., the gradual “bending” of the level sets is unstable during training, and some runs omit “bending” at all as it requires passing through the high loss value region).
>
>   **Action:** We will add this discussion to our Limitations section.
>
> - (3) Thank you for this correction. Indeed, by setting κ= 0 in our CSM, we do not achieve point identification. Generally, it is not clear how to generalize the point identification of BGMs, e.g., there is no natural notion of monotonicity for functions from R^d to R.
>
>   We summarized this result as a part of our *newly added theoretic result*, which we presented in Lemma 3 of the **PDF**. Therein, we introduce a so-called BGMs-EQTDs identification gap. This gap describes the closest we can get to the point identification by setting κ=0. Although our sensitivity model does not achieve full point identification for κ= 0, our CSM can still be useful for decision-making, as we show in our newly added case study (see **PDF**).
>
>   Regarding the computational experiment (Fig. 13):  when setting λ_κ=0 (same as κ= ∞), some runs omitted “bending” as that comes at the cost of the loss increase.
>
>   **Action:** We will add the new theoretical result from Lemma 3 (BGMs-EQTDs identification gap) to our revised paper.
>
> Minor Points
> * We will clarify throughout our paper that we work with ECOU, which are *expectations* of the counterfactual outcomes. Then, min/max are performed on top of the expectations.
> * Regarding Fig. 6, we have added an expanded version in the supplements of our original paper (see App. G, Fig. 12) with unidirectional edges, which explains the training and inference of APID.
>
> References:
> - [1] Celar, Lenart et al. "How people reason with counterfactual and causal explanations for AI decisions in familiar and unfamiliar domains." Memory & Cognition (2023): 1-16.
> - [2] Kevin Xia et al. “Neural causal models for counterfactual identification and estimation”. In: ICLR. 2023.
> - [3] Kevin Xia et al. “The causal-neural connection: expressiveness, learnability, and inference”. In: Advances in NeurIPS. 2021

---

> > ### Comment · Reviewer_8Zs8 · 2023-08-14
> >
> > Thank you for the detailed and helpful response - I'll give my quick reactions below
> >
> > (1), regarding the interpretation of $\kappa$ - I appreciate the intuition regarding the non-linearity between the outcome and latent noise that interacts with treatment.  From a mathematical / theoretician point of view, I think there is a certain elegance to that notion. However, I don't think that it helps much with practical application - I would imagine that the type of latent factors considered here (e.g., uniform random variables) often lack a clear real-world interpretation, making it difficult to use domain knowledge to choose bounds on non-linearity.
> >
> > (2), regarding the correspondence between the theoretical bounds & the bounds computed by the algorithm - I suspected as much, that there isn't a clear connection given the intractability of the original problem, and I take your response to confirm that suspicion.  I do appreciate your intuition regarding what goes wrong during optimization, e.g., "some runs omit 'bending' at all as it requires passing through the high loss value region)"
> >
> > (3), regarding the extremes of setting $\kappa$ to 0 or $\infty$:
> > * For the unconstrained case where $\kappa = \infty$, it does feel rather unsatisfying that we still have informative bounds, but this appears to be due (based on your responses) to the imperfect optimization procedure.
> > * For the highly-constrained case where $\kappa = 0$, am I correct to understand that the original statement was not entirely accurate?  E.g., on Line 366, BGM s referred to a special case of CSM with $\kappa = 0$, and on Line 72 in the intro it is stated that "we further show that we obtain the BGMs from [68] as a special case when setting the curvature to zero".  I appreciate the new theoretical result, but just wanted to clarify the context.

---

> > > ### Author Response · Authors · 2023-08-15
> > >
> > > Thanks for reading our response and we are again grateful for such an interest and attention to the details in our work.
> > >
> > > We would like to further clarify the abovementioned issues.
> > > - (1) Thank you for bringing up the issue of applicability. We argue that the real-life application of sensitivity models works in reverse: we rather try out different values of the sensitivity parameter and see what value changes our decision or conclusion. Let us consider the following use case of the marginal sensitivity model (MSM) for the smoking effect on lung cancer study. E.g., with the MSM it was shown that the odds of getting treatment (smoking) have to be almost 5 times higher conditioning on some confounder, to explain away the causal effect on lung cancer. MSM doesn't provide any information what is the interpretation or distribution of such a confounder, i.e., it is up to the domain expert to speculate about the existence of such a variable (or set of variables). In real life, the ground truth sensitivity parameter of the MSM is not known even for this smoking-lung cancer scenario.  The same could be said about the application of our CSM (see newly added Real-world case study). Regarding the uniformity of latent variables: considering the limited space of the NeurIPS submission, we only provided the most basic setting of our CSM when latent variables are assumed to be uniform and independent. We relied on this simplification for the sake of clarity and ease of the mathematical notation. The extension to arbitrary (absolutely) continuous variables and how non-linear transformations of those affect the curvature would be an interesting extension of our paper.
> > >
> > > - (2) Thank you for the nice summary, you understood it right. We would to like stress again, though, that our APID is an imperfect, but tractable heuristical instantiation of the CSM, which is a valuable contribution.
> > >
> > > - (3) The original statement was accurate but seemed misleading. CSM with $\kappa = 0$ indeed includes BGMs as a special case, but also the whole interval in-between. With Lemma 3 we also showed, that it additionally contains EQTDs and the interval in-between.
> > >
> > > We will add the discussion above to the revised version of the paper.

---

> > > > ### Comment · Reviewer_8Zs8 · 2023-08-15
> > > >
> > > > (1) Agreed that many sensitivity analyses proceed in this way, e.g., "how strong would a confounder have to be in order to over-turn our conclusions".  However, even judging the "plausibility" of a particular value requires some grounding in domain knowledge that might be difficult to elicit in this case.  E.g., perhaps a value of $\kappa = 5$ changes the result of some counterfactual inference - the natural next question would be whether or not this is a particularly "large" or "realistic" value.
> > > >
> > > > (2) Makes sense, thank you for the confirmation.
> > > >
> > > > (3) Understood, would be good to clarify in a revision.
> > > >
> > > > In the interests of moving "off the fence", I'm going to increase my score (borderline accept -> weak accept), in part due to the valuable and transparent discussion.

---

### Author Rebuttal · Authors · 2023-08-09

We are very grateful for the positive and in-depth feedback from the reviewers. We have carefully addressed all of the questions in the individual responses below. We will incorporate all changes (labeled with **Action**) into the camera-ready version of our paper.

We have additionally uploaded empirical results as a **PDF file**. Our main improvement is the following: we added a **new experiment with a real-world case study**. Therein, we adopt our curvature sensitivity model (CSM) to answer “what if” questions related to whether the lockdown was effective during the COVID-19 pandemic. We find that our CSM provides important and meaningful insights that are of practical value.

We want to highlight that, to the best of our knowledge, our paper is the first to address partial counterfactual inference for continuous outcomes. In particular, we are first to provide a measure-theoretic formulation to the partial identification task, and we are first to formulate the results of non-informative bounds for smooth functions. In general, designing a sensitivity model for counterfactual inference is a non-trivial task. For example, it is not clear how to straightforwardly generalize the point identification results for this purpose (e.g., there is no conventional notion of monotonicity for function from R^d to R). Our curvature sensitivity model (CSM) is one viable solution to build such a sensitivity model, where the sensitivity parameter $\kappa$ can be interpreted as the *level of non-linearity between the outcome and its latent noise variables that interact with treatment*.

---

### Decision · Program_Chairs · 2023-09-21

**Decision:**

Accept (spotlight)

**Comment:**

This paper contributes several new results for partial counterfactual identification of continuous outcomes. First, the authors show that if the class of SCMs is sufficiently smooth, then the ignorance interval of the expected counterfactual outcome of [un]treated becomes un-informative (Theorem 1). Second, the authors propose the Curvature Sensitivity Model which restricts the class of SCMs in order to obtain an informative bound. Finally, the authors propose a deep generative model to perform partial counterfactual identification under the proposed curvature sensitivity model.

All reviewers provide thoughtful and positive feedbacks regarding the novelty and the theoretical contributions of the paper. The authors also provide concrete action plans for the revision, notably to strengthen the numerical experiments of the paper. I strongly encourage the authors to implement thoroughly these action plans in the camera-ready version of the manuscript.